# A non-catalytic herpesviral protein reconfigures ERK-RSK signaling by targeting kinase docking systems in the host

Anita Alexa [1], Péter Sok [1], Fridolin Gross[2], Krisztián Albert[1], Evan Kobori[3], Ádám L. Póti[1], Gergő Gógl[1], Isabel Bento[4], Ersheng Kuang [5], Susan S. Taylor [6], Fanxiu Zhu [5], Andrea Ciliberto[2] & Attila Reményi [1✉]

The Kaposi's sarcoma associated herpesvirus protein ORF45 binds the extracellular signal-regulated kinase (ERK) and the p90 Ribosomal S6 kinase (RSK). ORF45 was shown to be a kinase activator in cells but a kinase inhibitor in vitro, and its effects on the ERK-RSK complex are unknown. Here, we demonstrate that ORF45 binds ERK and RSK using optimized linear binding motifs. The crystal structure of the ORF45-ERK2 complex shows how kinase docking motifs recognize the activated form of ERK. The crystal structure of the ORF45-RSK2 complex reveals an AGC kinase docking system, for which we provide evidence that it is functional in the host. We find that ORF45 manipulates ERK-RSK signaling by favoring the formation of a complex, in which activated kinases are better protected from phosphatases and docking motif-independent RSK substrate phosphorylation is selectively up-regulated. As such, our data suggest that ORF45 interferes with the natural design of kinase docking systems in the host.

[1] Biomolecular Interactions Research Group, Institute of Organic Chemistry, Research Center for Natural Sciences, H-1117 Budapest, Hungary. [2] IFOM, Istituto FIRC di Oncologia Molecolare, 20139 Milan, Italy. [3] Department of Chemistry and Biochemistry, University of California San Diego, San Diego, 9500 Gilman Drive, La Jolla, CA 92093-0654, USA. [4] European Molecular Biology Laboratory, Hamburg, Germany. [5] Department of Biological Science, Florida State University, Tallahassee, FL 32306-4370, USA. [6] Department of Pharmacology, University of California San Diego, 9500 Gilman Drive, La Jolla, San Diego, CA 92093-0654, USA. ✉email: remenyi.attila@ttk.hu

As obligate intracellular pathogens, viruses affect host signaling pathways to alter the signal strength, duration, and signaling specificity to their advantages. One of the most direct mechanisms relies on the genetic usurpation of host enzymatic components (e.g., *v-ras* GTPase or *v-src* protein kinase from different sarcoma viruses)[1,2], but non-catalytic viral proteins also play an important role in reconfiguring intracellular signaling, albeit more indirectly, by affecting protein-protein interactions[3]. Many viruses target mitogen-activated protein kinase (MAPK) pathways because of their critical role in controlling cell growth[4,5]. The non-catalytic ORF45 protein of Kaposi's sarcoma-associated herpesvirus (KSHV) causes sustained activation of extracellular signal-regulated kinase (ERK1/2) and p90 ribosomal S6 kinase (RSK1/2) by forming a complex with the two kinases[6,7]. The N-terminal ORF45 region (~80 aa) has been shown to be required and sufficient for ERK and RSK activation. In particular, F66 located in a short relatively conserved region, referred to as VF motif henceforth, has been identified as the most critical for binding to and activation of RSK. Accordingly, an ORF45-F66A mutant virus failed to cause sustained ERK and RSK activation during lytic reactivation and had produced less infectious progeny viruses in comparison to the wild type or the revertant[8,9]. Similarly to how catalytic viral proteins had helped the blueprinting of signaling pathways several decades ago[10], non-catalytic viral proteins may help to reveal the molecular logic based on which complex signaling networks operate.

ERK and RSK are key downstream components of the epidermal growth factor (EGF)/Ras/ERK pathway. RSK is one of the downstream substrates of ERK and the two proteins are known to form a heterodimer[11]. RSK consists of two kinase domains connected by a flexible linker[12,13]. The N-terminal AGC-type kinase domain (NTK) phosphorylates downstream substrates after it is activated by phosphoinositide dependent kinase 1 (PDK1) through activation loop phosphorylation. PDK1 is recruited to RSK after hydrophobic motif (HM) phosphorylation which is known to be mediated by the C-terminal calcium/calmodulin-dependent kinase (CAMK) domain (CTK)[14]. Activated ERK phosphorylates the CTK at its activation loop and binds to RSK at its C-terminal tail containing a MAPK binding linear D(ocking)-motif[11,15]. Serine/threonine kinases often rely on docking motifs, which are short linear binding motifs located in the disordered part of their binding partners, to interact with their substrates and with modifying enzymes (e.g., activators or deactivators). Conversely, docking grooves on kinases enable more efficient and specific phosphorylation of substrates but they are also engaged in binding to upstream kinases and phosphatases[16,17]. This competition requires the exploration of the underlying protein network topology and its parameters.

To understand how ORF45 forms a complex with both ERK and RSK and causes persistent activation of the two kinases, we determined the crystal structures of ppERK2-ORF45 and RSK(NTK)-ORF45 binary complexes. We show that ORF45 binds to ERK and to the RSK NTK using two distinct linear binding motifs, while RSK engages with ERK using the D-motif located at the tail of the RSK CTK. The structural model of the ternary complex illustrated that ORF45 holds the three kinase domain-containing ERK2-RSK2 complex together by docking motif-mediated interactions. ERK-RSK interaction is more stable in the ternary ERK-RSK-ORF45 complex and the kinases are better protected from dephosphorylation by phosphatases. Moreover, our work highlights kinase docking grooves that are hijacked by pathogens and enable them to reconfigure ERK-RSK MAPK signaling leading to RSK substrate phosphorylation. Successful reconfiguration requires an optimized ability to bind other proteins and goes beyond simple interference with binary binding: it is a context-dependent perfection in the host's full kinase docking system including activators, phosphatases and substrates. In conclusion, KSHV ORF45 is a systems level manipulator of protein-protein binding and relies on perfected kinase docking motifs to manipulate ERK-RSK signaling.

## Results

**ORF45 enhances EGF-induced ERK and RSK activation**. In order to elucidate the effect of ORF45 on the ERK-RSK signaling pathway, we created a HEK293T cell line allowing doxycycline inducible expression of this viral protein (HT-ORF45; HEK293-Tet-ON). Expression of ORF45 increased the levels and durations of ERK and RSK phosphorylation upon EGF treatment, confirming that ORF45 causes persistent activation of ERK and RSK in cells[6,7] (Fig. 1a). We also examined the effect of ORF45 on ERK signaling in the RSK1/2 knock-out cell line (HEK293T-ΔRSK1/2)[18] and found that ORF45 could not activate ERK in the absence of RSK (Fig. 1b). These results showed that the effect of the viral protein on ERK was dependent on one of its downstream substrates, RSK.

We next investigated the impact of ORF45 on ERK-RSK interaction in cells using a luciferase complementation-based protein–protein interaction (PPI) assay. ERK2 and RSK2 were tagged with either half of the firefly luciferase enzyme and the reconstituted luminescence signal was monitored in cells in the presence and absence of ORF45. We also tagged p38 and MK2, two paralogous kinases forming the p38-MK2 complex, and results indicated that ORF45 selectively increased ERK2-RSK2 interaction but not p38-MK2 interaction (Supplementary Fig. 1). To identify RSK2 regions instrumental for ORF45-mediated increase in ERK2-RSK2 binding, we repeated the experiment with various truncated RSK2 constructs (lacking the C-terminal tail containing D motif, the CTK, or the NTK) (Fig. 1c). As expected, the C-terminal tail is required for ERK2-RSK2 binding because deletion of the tail from either WT or CTK abolished this interaction[11]. We detected strong ORF45-mediated increase for ERK-RSK binding if the NTK was part of the RSK construct, suggesting that ORF45 interacts with the NTK.

**ORF45, RSK2, and ERK2 form a ternary complex**. In order to elucidate the binding of the N-terminal ORF45 region to RSK and/or ERK, we labeled a 61 amino acid long N-terminal peptide with a fluorescent dye and used size-exclusion chromatography to show that ORF45(16-76) binds to RSK2 as well as to the ERK2-RSK2 complex (Supplementary Fig. 2a). Furthermore, small X-ray scattering (SAXS) analysis of RSK2 alone, the binary ERK2-RSK, and the ERK2-RSK2-ORF45 ternary complex showed that the proteins form a 1:1:1 stoichiometric complex, and dimensionless Kratky-plots indicated that the ternary complex has a compact domain arrangement (Supplementary Figs. 2b and 3a–c).

To explore the PPIs in the ERK2-RSK2-ORF45 ternary complex, we used the ORF45(16-76) peptide and examined its binding to ERK2 and RSK2 in vitro. The N-terminal ORF45 region contains an FxFP-motif. Such motifs are present in several ERK substrates and are known to bind at the so-called F-site with higher affinity if the kinase is double-phosphorylated (ppERK2)[19]. ORF45(16-76) was found to bind to ppERK2 with <1 μM affinity, and a shortened version (16-40) retaining the FxFP sequence bound similarly. Importantly, we found that ORF45(16-40) interacts with ppERK2 at least 20-fold stronger than with nonphosphorylated ERK2 (Fig. 2a and Supplementary Fig. 4a). To reveal the structural basis of this interaction, we determined the crystal structure of the ppERK2-ORF45(27-40) complex (Supplementary Table 1). This crystal structure of a

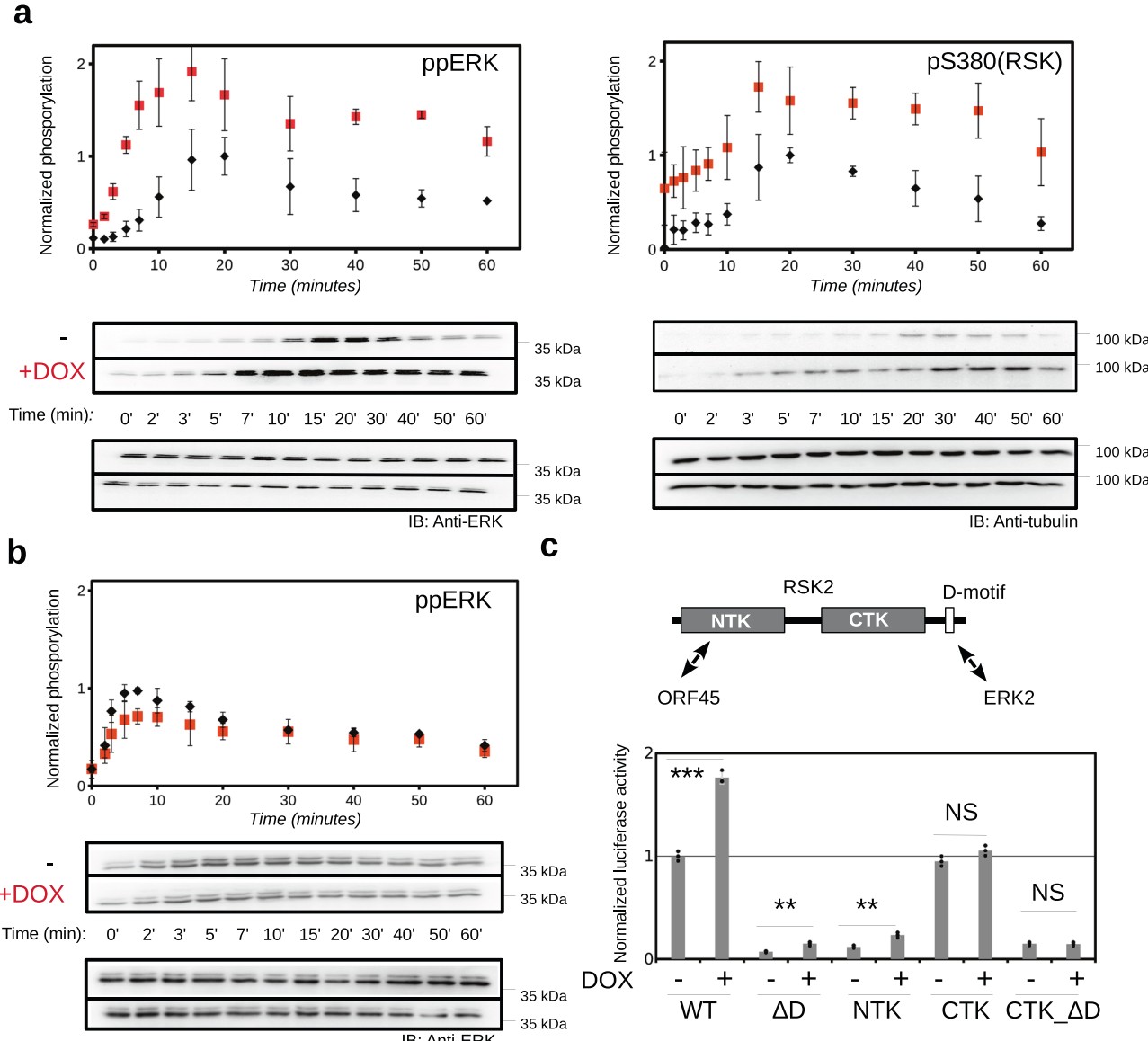

**Fig. 1 EGF triggered ERK and RSK activation in the presence of ORF45.** HEK293T (**a**) and HEK293T-ΔRSK1/2 (**b**) cells ('HT-ORF45') were treated with 100 ng/mL EGF and ERK (ppERK) and RSK (pS380) phosphorylation was monitored in Western-blots. Doxycycline treatment (DOX) was used to induce ORF45 expression (data for untreated and DOX treated cells are shown in black or red, respectively). Anti-panERK antibody was used as the load control for ppERK, and anti-tubulin antibody for pRSK Western-blots. Anti-phospho Western-blot signals were normalized to the Control signal (-: no DOX treatment) at 20 min for HEK293T cells and to the 7 min Control signal for HEK293T-ΔRSK1/2 cells. Error bars indicate SD of three independent experiments. Data points show the mean ± SD (N = 3). Western-blot panels show a representative set. **c** Mapping of RSK2 regions involved in the ORF45 mediated increase of ERK2-RSK2 binding. Different RSK2 constructs and ERK2 were tagged with the two different fragments of the split firefly luciferase enzyme and luminescence was normalized to the signal of the ERK2-RSK2 interaction (WT). WT: full-length RSK2, ΔD: RSK2 construct lacking the C-terminal tail containing the D-motif. Error bars show SD based on three independent experiments. (N = 3; Paired t-test, two-sided; NS not significant, *p < 0.05, **p < 0.01, ***p < 0.001). Source data are provided as Source data file.

ppERK2-FxFP-motif complex showed that the F-site is outlined by hydrophobic residues (I198, M199, L200, Y233, L234, L237, and Y263), and comparison to the non-phosphorylated ERK2 structure[20] reveals that the bulky phenylalanine side-chains and the proline from the FxFP-motif are better accommodated if the ERK2 activation loop is phosphorylated (Fig. 2b).

Next, we mapped RSK2-ORF45 binding. First, the binding of RSK2 to ORF45(16-76) was investigated by using surface plasmon resonance (SPR). RSK2 was expressed biotinylated and immobilized to the streptavidin surface of the SPR chip. The binding affinity was found to be ~1 nM, which was more than

1000-fold higher than that of the ppERK2-ORF45 binary complex. ORF45 binds to the full-length RSK2 or NTK with similar $K_D$ suggesting that the NTK alone is sufficient for high-affinity binding (Supplementary Fig. 4b). In contrast, a short region, ORF45(60-70), containing F66 from the VF-motif, bound to the NTK with only micromolar affinity ($K_D \sim 20 \mu M$), but a longer ORF45(56-76) peptide bound stronger ($K_D \sim 2 \mu M$), indicating that the binding region extends beyond the core VF-motif (Supplementary Fig. 4c).

To reveal the structural basis of NTK-ORF45 binding, we determined the crystal structure of the NTK-ORF45(16-76) complex at 2.75 Å resolution (Supplementary Table 1). The asymmetric unit

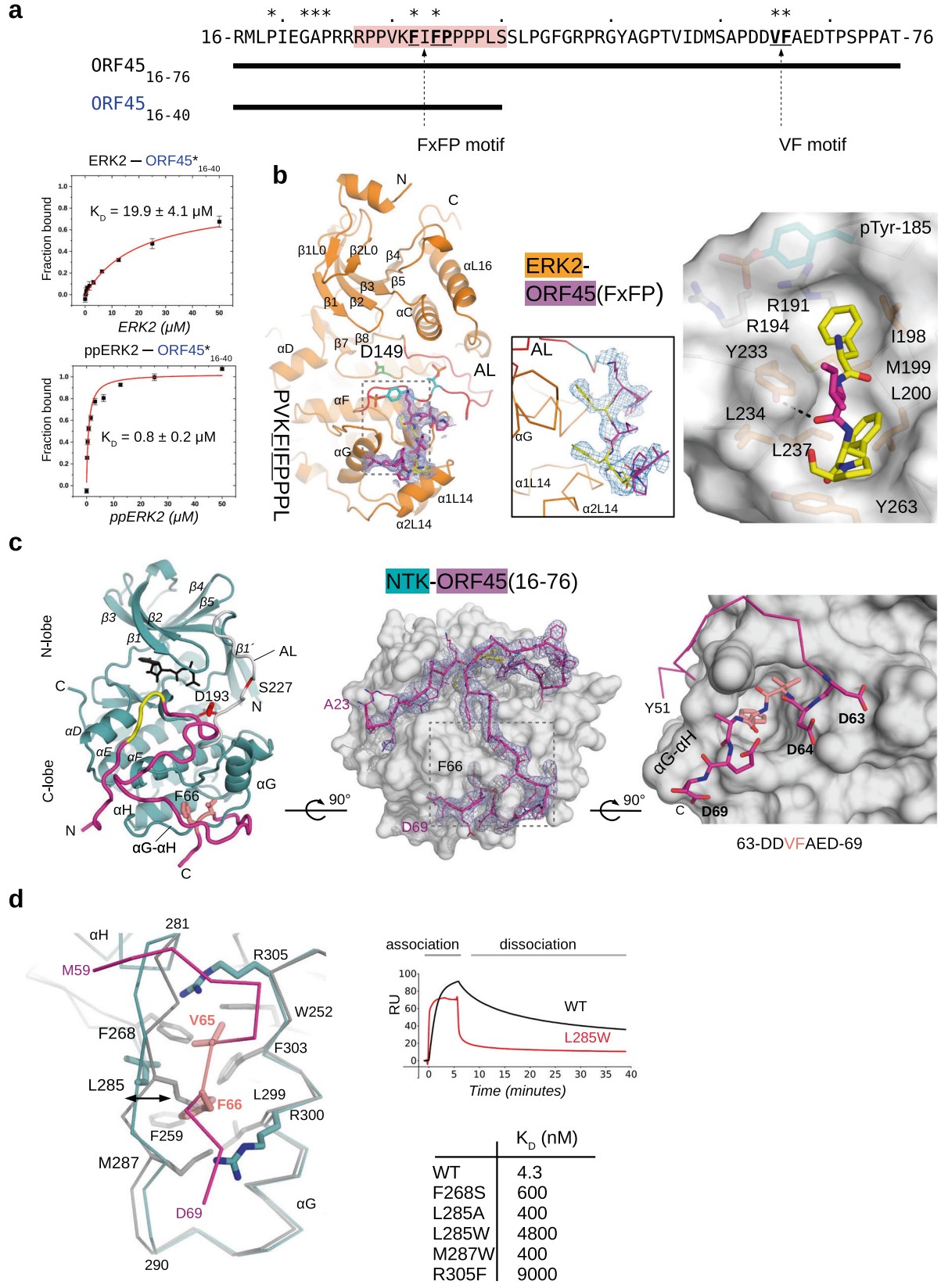

**a**

16-RMLPIEGAPRRRPPVK**FIFP**PPPLSSLPGFGRPRGYAGPTVIDMSAPDD**VF**AEDTPSPPAT-76

ORF45 16-76

ORF45 16-40

FxFP motif

VF motif

**b** ERK2-ORF45(FxFP)

**c** NTK-ORF45(16-76)

63-DD VF AED-69

**d**

| | $K_D$ (nM) |
|---|---|
| WT | 4.3 |
| F268S | 600 |
| L285A | 400 |
| L285W | 4800 |
| M287W | 400 |
| R305F | 9000 |

contained 6 NTK-ORF45 complexes and 47 amino acids from ORF45 could be traced in the electron density map. ORF45 makes contacts with the C-lobe of the NTK (Fig. 2c). The N-terminal FxFP motif is located between the N- and C-lobe but the most prominent contacts are formed by the C-terminal VF-motif containing region: V65 and F66 are deeply buried in a hydrophobic pocket between the αG-αH loop and helix G (outlined by W252, F259, F268, L285, M287, L299, F303 and also involving R300 and 305), while D63-64 and D69 make H-bonds or salt bridges with K304, N309, and R300, respectively. The ORF45 contact surface on the NTK was validated by mutating critical residues at this region. As expected, binding affinity

**Fig. 2 Crystal structure of the ppERK2-ORF45 and RSK2 NTK-ORF45(16-76) complex. a** The N-terminal ORF45(16-76) region has an FXFP motif. Evolutionarily conserved positions—among mammalian gamma herpesviruses—are marked with an asterisk. Panels show results of fluorescence polarization-based protein–peptide binding assays using the carboxyfluoresceine (CF) labeled (*) ORF45(16-40) peptide. (ppERK2 —double-phosphorylated ERK2). Error bars show SD based on three independent experiments. Data points show the mean ± SD. **b** Crystal structure of the ppERK2-ORF45(27-40) complex. The left panel shows ppERK2 in cartoon (orange) and a 10 amino acids long region containing the FxFP motif (in magenta, where underlined amino acids are colored in yellow). The figure shows the $\sigma_A$-weighted simulated annealing omit map contoured at 1.5 and a 10 amino acids long region containing the FxFP motif (in magenta, where underlined amino acids are colored in yellow). The figure on the right shows a zoomed-in view of F-site residues and the FxFP motif peptide. The phosphotyrosine residue of the activation loop (pTyr-185; coordinated by R191 and R194) forms the upper part of the F-site pocket. **c** Crystal structure of the RSK NTK-ORF45(16-76) complex. ORF45 (in magenta) binds to the C-lobe of NTK (in teal) with extensive contacts. The panel in the middle shows the NTK from the bottom: the NTK is shown with surface representation and the ORF45 peptide is shown with $\sigma_A$-weighted simulated annealing Fo-Fc omit map contoured at 2σ. The panel on the right shows a zoomed-in view of the VF motif binding region. The N-terminal FxFP motif (in yellow) is located in the broad crevice between the N- and C-lobe, while the C-terminal VF motif (in salmon) binds to a small hydrophobic pocket by αG/αH and the intervening αG-αH loop. D193 from the active site and the PDK phosphorylation site (S227) in the activation loop (AL) are highlighted in the panel on left. **d** Comparison of the NTK structure in the ORF45 bound complex versus in the apo NTK structure (PDB ID: 4NW6, shown in gray) by the VF-motif binding region. The side-chain of L285 is flipped-out so that to make room for F66 from ORF45. Notably, the hydrophobic pocket widens (indicated by an arrow) because region 281-290 (αG-αH loop) changes its conformation. The two panels in the middle show the comparison of SPR kinetic binding curves for wild-type (WT) NTK and the L28W mutant (top)—injected at a concentration corresponding to the $K_D$— and the summary of binding affinity measurements ($K_D$—equilibrium dissociation constant) with WT NTK or VF-motif binding surface mutants (bottom) (see Supplementary Fig. 4d). Source data are provided as Source data file.

decreased by ~100-1000-fold when F268 was changed to serine or L285, M287 or R305 were replaced with alanine or with bulkier tryptophan/phenylalanine residues (Supplementary Fig. 4d). The binding of the N-terminal ORF45 segment to the RSK NTK was unexpected since this region is known to bind to activated ERK with its FxFP motif (and binds ERK in the ERK-RSK-ORF45 ternary complex). The fact that this N-terminal ORF45 region is also visible in the RSK(NTK)-ORF45(16-76) binary crystal structure is consistent with the increased binding affinity of longer ORF45 constructs. Comparison of the NTK surface from the RSK2 NTK-ORF45 complex with that from apo-NTK structures[21–23] revealed that the hydrophobic pocket accepting F66(ORF45) widens: L285 is flipped-out and the loop connecting αG and αH (281-290) changes its conformation (Fig. 2d). SPR can be used to study the kinetics of binding: ORF45 bound to NTK(WT) with slower association (on-rate) and a lot slower dissociation (off-rate) compared to the NTK(L285W) mutant, while the latter displayed both fast association and dissociation. Fast on- and off-rates are typical when peptides bind to a rigid protein surface, as this is possibly the case for the L285W mutant where only the N-terminal ORF45 region could be involved in binding since the F66(ORF45) binding slot on the NTK is disrupted. The NTK(WT)-ORF45 interaction has a slow on-rate because of the structural rearrangement that needs to take place before VF-motif binding since L285 is incompatible with F66(ORF45) in its original position. Once the VF-motif is docked into the rearranged binding slot, the off-rate will drop since the new interactions "seal" the complex[24].

**Structure of the ERK-RSK-ORF45 ternary complex.** Hydrogen Deuterium Exchange Mass Spectrometry (HDX-MS) experiments on the ERK-RSK binary and ERK-RSK-ORF45 ternary complexes showed that the αG-αH loop is strongly protected in the presence of ORF45. This is in agreement with the crystal structure of the NTK-ORF45 binary complex because binding of ORF45 to the NTK in the ternary complex would directly protect this part of RSK2. In addition, the RSK2 region forming the contact between ERK and the CTK—namely, the extended αF helix with the APE motif[11]—is also better protected indicating that the ERK-CTK interface is stabilized by ORF45 binding (Fig. 3a and Supplementary Table 2).

The crystallographic models of the ppERK2-ORF45, NTK-ORF45, and CTK-ERK2 (PDB ID: 4NIF) complexes were used to create the structural model of the ternary complex based on SAXS

data collected on the ERK2-RSK-ORF45(16-76) sample (Fig. 3b). Solution scattering data matched well to a ternary complex model in which ERK and CTK are engaged in a tight complex as in their crystal structure, and the NTK-ORF45 and ERK-ORF45 crystallographic models are flexibly linked through the ORF45 intervening segment between the FxFP- and VF-motif containing regions. Overall, the model reveals how the two binding motifs in ORF45 and the D-motif in RSK hold the three kinase domains in a compact structure—in spite of the long and flexible intervening region between the two kinase domains of RSK—where the NTK active site is open and accessible (Fig. 3c).

In order to validate the binary PPI contacts and the effects of ORF45 on these, we used a cell-based dynamic luciferase complementation assay (NanoBit)[25,26]. First the impact of VF and FxFP region mutations were tested on ORF45-RSK2 binding in live cells. Mutating the phenylalanine in the VF-motif for alanine (F66A) greatly decreased luciferase enzyme complementation, while FxFP-motif mutations did not affect this binary interaction, as expected (Fig. 3d). Next, the luminescence of the same RSK2-ORF45 probes were measured after EGF treatment, where the WT signal remained unchanged but the VF-motif mutated pair displayed a two-fold increase upon EGF treatment. Similar to this, ERK2-ORF45 interaction probes showed a greater increase for the VF mutated ERK2-ORF45(F66A) pair (~40-fold vs ~3-fold for WT). These data suggest that activated ERK2 binds to ORF45 through its FxFP-motif with increased affinity in cells and that ERK2-RSK2-ORF45 forms a more stable and thus less dynamic ternary complex if the ORF45-RSK2(NTK) interaction is intact.

**Kinetics of ERK2-RSK2-ORF45 ternary complex assembly in vitro.** SPR can be used to monitor protein–protein binding in real-time and gives information on the kinetic binding rates as well as on the $K_D$ ($K_D = k_{off}/k_{on}$). The RSK2-ORF45 complex was found to have a small $k_{off}$ in contrast to the fast-dissociating ppERK2-RSK2 complex (Fig. 4a). In further SPR experiments we monitored binding of ERK2 to RSK2 in the presence of ORF45. These experiments showed that the ternary complex is more stable than the nonphosphorylated ERK2-RSK2-ORF45 complex (Fig. 4b).

We created a quantitative mechanistic model of the in vitro systems studied by SPR. We based our model on the binary binding experiments and obtained the binding rate constants that described the binary interactions between ERK2, RSK2, or ORF45

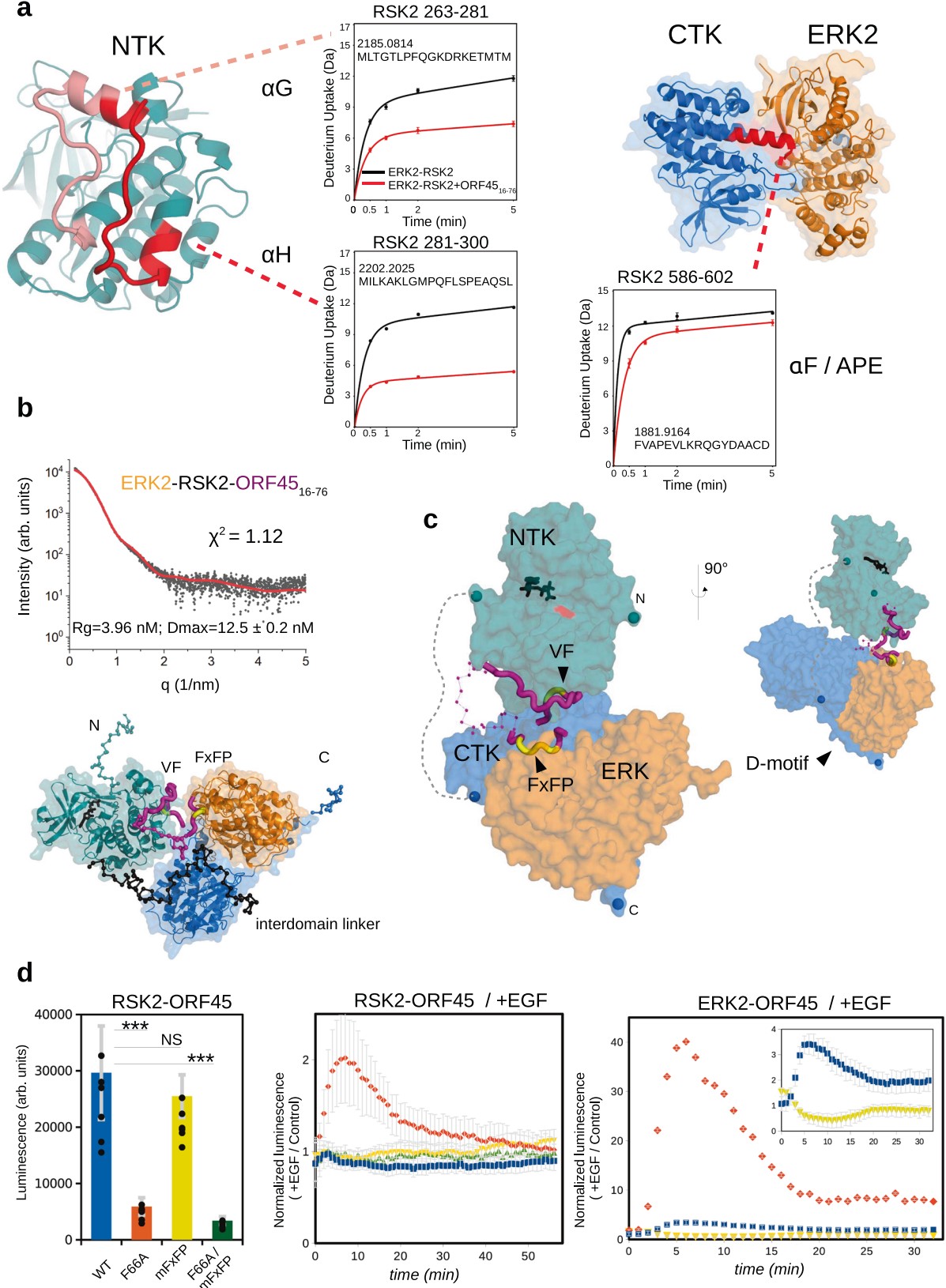

($k_{on}$ and $k_{off}$-s; ERK2-RSK2, ERK2-ORF45, and RSK2-ORF45) (referred to as the Biophysical Model, see later). However, using the optimized parameters that fitted all binary binding experimental data well, the SPR binding curves for the ppERK2-RSK2-ORF45 ternary samples could not be reproduced, suggesting that ORF45 and ERK binding to RSK2 are likely not independent

(Supplementary Fig. 5). We posited that the kinetic constants determined in the absence of the third protein do not describe the binding events in the ternary complex appropriately. For example, the "closed" complex in which all three components bind to each other is likely more stable than any of the possible three "open" complexes in which only two proteins form protein-

**Fig. 3 Structural model of the ERK2-RSK2-ORF45 complex and PPI validation in cells. a** Hydrogen Deuterium Exchange Mass Spectometry (HDX-MS) of the RSK2-ERK2 binary complex (in black) and RSK2-ERK2-ORF45 ternary complex (in red). Deuterium uptake of selected peptides are displayed onto the NTK from the NTK-ORF45 crystal structure or onto the CTK-ERK2 crystal structure (PDB: 4NIF). Data are represented as mean values ± SEM of three technical replicates. **b** Results of the SAXS analysis and the CORAL model of the ternary complex. The panel on top shows the experimental SAXS curve (black dots) with the simulated curve (in red) calculated based on the structural model shown on the panel below. Flexible N- and C-terminal elements (N or C) of RSK, the interdomain linker of RSK (colored in black), and the intervening region of the ORF45 peptide between the VF and FxFP motif regions are shown with thin ribbon and their Cα atoms are shown with spheres. The structural model was built using three binary crystallographic models (ppERK2-ORF45, NTK-ORF45, and CTK-ERK). **c** The ppERK2-RSK2-ORF45(16-76) ternary complex shown from two different orientations. The ORF45 is colored in magenta, the core FxFP and VF motifs are shown in yellow, and the intervening region between these two is shown with thin ribbon where Cα atoms are shown with spheres. The flexible interdomain linker between the NTK and CTK is indicated with a dashed line. N and C denotes N- and C-terminal residues of RSK where their Cα atoms are shown with spheres. The AMPPNP in the NTK nucleotide pocket is shown with black sticks and the active site residue (D193) is colored red. **d** Monitoring binary RSK2-ORF45 and ERK2-ORF45 binding in cells. The panel on the left shows the NanoBit luciferase signal for ORF45-RSK2 binary interaction. Middle panel: Normalized NanoBit signal for the ORF45-RSK2 interaction probes upon EGF treatment. Right panel: Normalized NanoBit signal for the ORF45-ERK2 interaction probes upon EGF treatment. The inset shows the same plot but without the curve for the ORF45(F66A)-ERK2 probe. WT (blue): full-length ORF45; F66A (red): F66 from the VF motif replaced to alanine; mFxFP (yellow): FxFP region is mutated to 4 alanines; F66A/mFxFP (green): both regions are mutated. The luminescence signal after EGF treatment was normalized to the signal from untreated cells (control). Error bars in light gray show SD based on three or six independent experiments. Data points show the mean ± SD. (N = 6, left panel; N = 3, middle and right panel; Paired t-test, two-sided; NS not significant, *p < 0.05, **p < 0.01, ***p < 0.001). Source data are provided as Source data file.

protein contacts (Fig. 4c). Therefore, we introduced two global correction factors ("a" and "d" affecting association and dissociation, respectively) assuming that the association and dissociation rates relevant for the closed ternary complex are proportional to the corresponding rates of the binary reactions. These two values were left free in the fitting procedure and the SPR data could be fitted only when "d" was <1 (d = 0.0027) and "a" also had a small value (a = 0.087 μM). This indicates that the apparent dissociation rate in the closed ternary complex is greatly diminished compared to the corresponding rate in the binary complex, but the apparent association decreases only to a lesser extent at a physiologically relevant concentration (~1 μM). These values of "a" and "d" therefore suggest that the closed complex is stabilized compared to the open ternary complexes. Formally, this can be described as an allosteric process—in a general sense where two binding events affect each other: VF-motif binding to RSK affects the binding of the RSK D-motif as well as the binding of the FxFP-motif to ERK. Since the involved sites are not on one protein but are located on different proteins of the ERK-RSK complex, it may be referred to as "complex" allostery.

**Substrate phosphorylation and ERK/RSK dephosphorylation.** Earlier experiments showed that ORF45 promotes activation of both ERK and RSK in cells. To elucidate the biochemical basis of this positive effect on these two kinases, we performed kinase assays using purified proteins. The impact of different N-terminal ORF45 fragments were tested on the ppERK2 → RSK2 as well as on the MKK1 → ERK2 → RSK2 reaction (Fig. 5a). Unexpectedly, ORF45(16-76) blocked RSK2 phosphorylation and shorter fragments also exhibited some degree of inhibition. These results argue against a mechanistic model that postulates a direct role of ORF45, for example by a classical allosteric mechanism, in kinase activation.

Next, RSK2 substrate phosphorylation was tested in similar in vitro kinase assays but ORF45 was added after RSK2 was pre-activated by ppERK2 and PDK1 (Fig. 5b). One of the best-established RSK substrates is translation initiation factor 4B (EIF4B)[27,28]. A short region of this substrate protein— known to be regulated by RSK at S422— was produced as GST-fusion protein and was used as an RSK substrate. As expected, efficient phosphorylation of this RSK NTK substrate required active PDK1 in addition to activated ERK2. More importantly, ORF45 did not interfere with NTK mediated phosphorylation of its substrate, suggesting that once the NTK is active ORF45 binding does not block RSK activity. Briefly, ORF45 blocks

ERK → RSK phosphorylation but it does not affect RSK → substrate phosphorylation in vitro.

We reason that ORF45 inhibits MKK1 → ERK2 → RSK2 signaling in vitro because it stabilizes the ERK-RSK binary complex in which the MAPK docking groove is occupied by the C-terminal tail of RSK. This could prevent MKK1—also depending on docking to ERK—from accessing its substrate. If ORF45 cannot activate but rather inhibits ERK activation, how could ORF45 cause elevated ERK2 and RSK phosphorylation in cells? In cells, phosphatases counteract the action of kinases. In order to address the role of phosphatases, we tested the effect of MKP3 which is a known phosphatase involved in ERK2 dephosphorylation. MKP3 requires MAPK docking groove binding for efficient MAPK dephosphorylation[29]. RSK alone did not protect ppERK2 from dephosphorylation, presumably because of the dynamic nature of this interaction (since it has a high $k_{off}$, see Fig. 4a), but ORF45, particularly in combination with RSK2, was efficient in protecting ERK2 from the phosphatase (Fig. 5c). These findings could be explained by the following: (1) FXFP-motif binding directly protects the ERK2 activation loop from dephosphorylation as the former shields the latter from the active site of the phosphatase; and (2) the higher stability of the ERK2-RSK2 complex in the presence ORF45 indirectly protects ERK2 from dephosphorylation because the C-terminal tail of RSK shields the ERK2 docking groove from the binding of the phosphatase.

**Modeling the impact of ORF45 on ERK signaling.** ORF45 inhibits ERK → RSK phosphorylation in vitro but it is an efficient activator of ERK-RSK signaling in cells. We were intrigued by this discrepancy and given the presence of many interacting molecules we decided to investigate the possible impact of ORF45 on the ERK signaling pathway by mathematical modeling. MAPKs bind to their activator kinases, substrates and phosphatases using dedicated PPI surfaces: the D-motif binding D(ocking)-site and the FxFP-motif binding F-site[30]. MKKs (MAPK kinases), RSK, and MKPs (MAPK phosphatases) all bind at the D-site and their interaction is mutually exclusive[31,32]. Furthermore, some substrates and ORF45 bind to the F-site. F-site mediated interactions are also central in the action of phosphatases: the phosphorylated MAPK activation loop— which is the target site for the phosphatase —is occluded and protected from dephosphorylation by FxFP-motif binding (see Fig. 2b). Because of this complex relationship between MAPKs and their interacting proteins, understanding how ORF45 affects ERK signaling is not straightforward. We created a quantitative in silico model based on in vitro

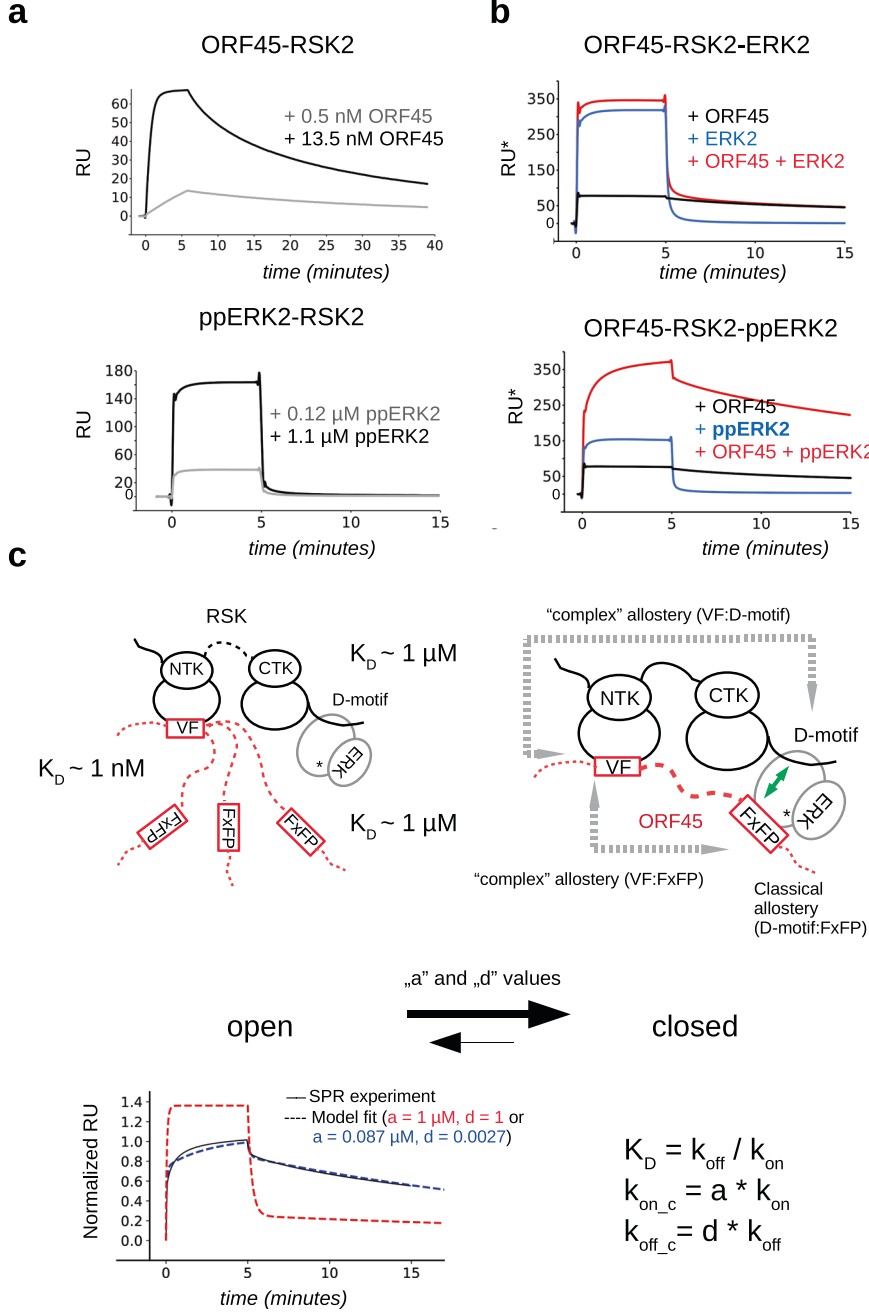

**Fig. 4 Characterization of ORF45-RSK2 binding and its effect on ppERK2-RSK2 binding. a** Binding between RSK2 and ORF45 or ppERK2 analyzed by SPR. ORF45 or ppERK2 were injected over the immobilized RSK2 surface in two different concentrations for 5 min (association phase) and then the dissociation of the complex was monitored in time. RU response units. **b** Comparison of ternary complex formation with ERK2 or ppERK2. Results of kinetic binding experiment using nonphosphorylated ERK2 (upper panel) or ppERK2 (lower panel) where proteins were injected over the immobilized RSK2 surface alone (10 µM ORF45 in black or 1 µM ERK2 in blue) or together (in red). **c** Schematics of the ppERK2-RSK2-ORF45(16-76) ternary complex. The "closed" ternary complex, which could form from three different "open" complexes (but only one of them is shown on this panel), is formed by three linear motif (VF, FxFP, and D-motif) mediated protein-protein contacts. The binding affinity of the binary interactions between VF(ORF45):NTK(RSK2), FxFP(ORF45):ppERK2, and D-motif(RSK2):ppERK2 are indicated. Note that VF:NTK binding is ~1000-fold stronger and has low $k_{off}$ compared to the other two interactions. Classical vs "complex" allostery within the ternary complex are highlighted with green or gray arrows, respectively. In the simulation of SPR data, the difference between binding rates in the binary versus the ternary complex could be introduced by the global "a" and "d" values affecting the $k_{on}$ or the $k_{off}$ of complex formation, respectively. The lower panel shows the simulation results (dashed lines) overlaid with the corresponding experimental binding curve (solid black line) where the ternary $k_{on}$ and $k_{off}$ are the same as in the binary complexes ($a = 1$ and $d = 1$; red) or adjusted by the fitted a and d values (blue). The apparent kinetic binding parameters in the closed complex ($k_{on\_c}$ or $k_{off\_c}$) are obtained by the multiplication of the binary binding constants by $a$ or $d$. Source data are provided as Source data file.

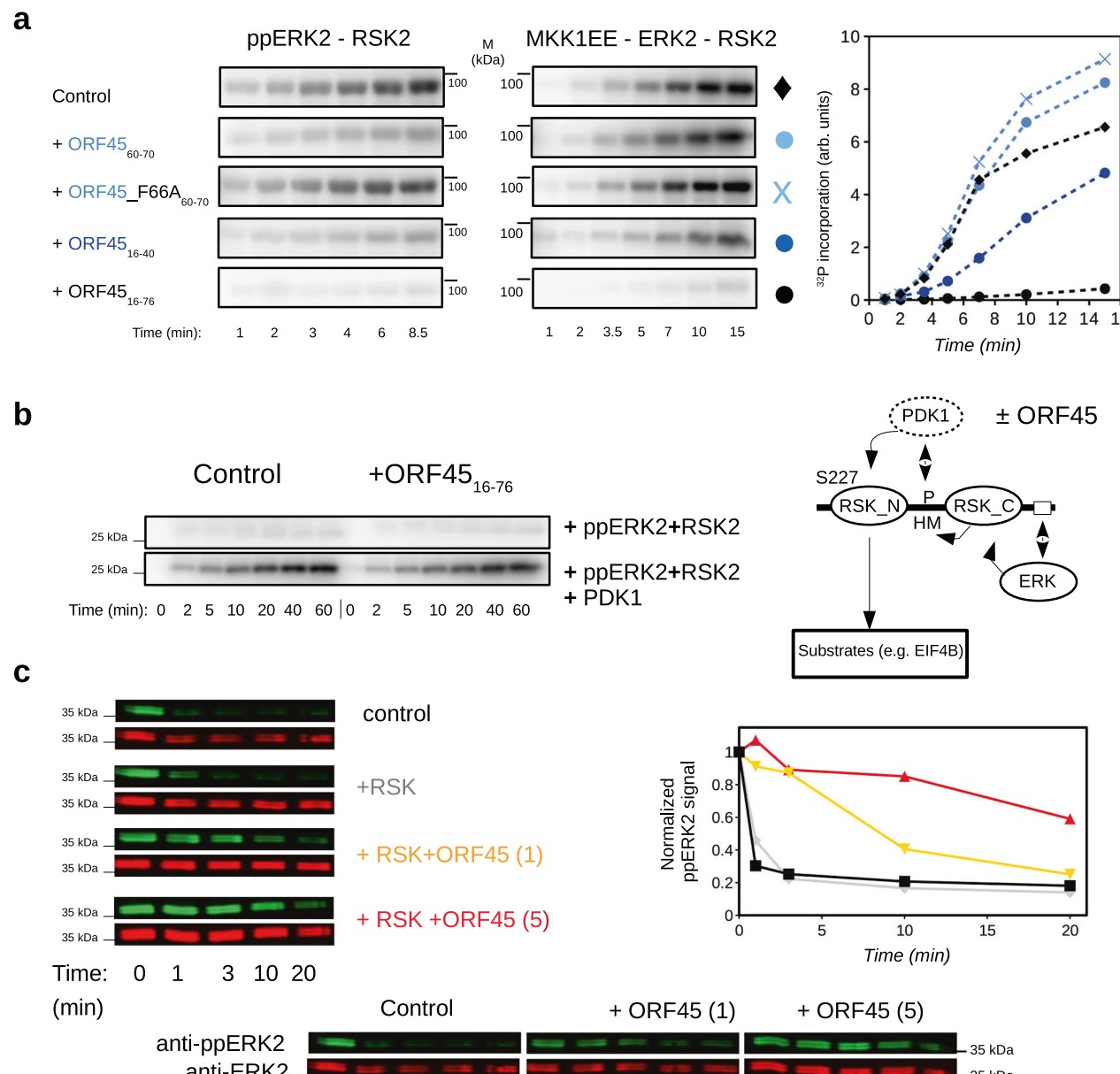

**Fig. 5 RSK substrate phosphorylation and ERK/RSK dephosphorylation ± ORF45. a** In vitro kinase assay results of RSK2 phosphorylation. Left panel: 10 nM ppERK2 was used to phosphorylate RSK2 (2 μM) and RSK2 phosphorylation was monitored at different time points by phosphorimaging in the absence (Control) and the presence of ORF45 peptides (50 μM). Middle panel: 0.25 μM activated MKK1 (MKK1EE) was mixed with 2 μM nonphosphorylated ERK2 and 2 μM RSK2; and RSK2 phosphorylation was monitored similarly as on the left panel. Phosphorimaging results are plotted on the panel at the right based on one experiment ($N = 1$). **b** In vitro kinase assay results on RSK substrate phosphorylation. 0.5 μM ppERK2 and 1 μM RSK2 was pre-incubated for 1 h, then 10 μM PDK1 was added and the phosphorylation of 10 μM GST-4EIFBpep—an RSK model substrate—was monitored in the absence (Control) or the presence of 10 μM ORF45(16-76) ($N = 1$). The schematic panel on the right highlights the hierarchical phosphorylation events (one-end arrows) and the PPIs (two-end arrows) required for NTK mediated phosphorylation of RSK substrates. (HM hydrophobic motif). **c** Results of ppERK2 dephosphorylation assays in the presence of RSK2 and ORF45. Control: 1 μM ppERK2 was incubated with 1 μM purified MKP3 and dephosphorylation was monitored by quantitative Western-blots (green signal: anti-phosphoERK2; red signal: anti-ERK2). Lower panels show the results of the same dephosphorylation assay but in the presence of 1 μM RSK2 and 1 or 5 μM ORF45(16-76). Panels at the bottom show results of the same assay but containing ORF45(16-76) without RSK ($N = 1$). Source data are provided as Source data file.

biochemical experimental data (Biochemical Model: an extension of the Biophysical Model on ERK-RSK-ORF45), and this was then extended and fine-tuned based on in-cell measurements finally leading to the more complex Network Model (Fig. 6a).

The initial binding parameters of the Biochemical Model were based on our former measurements. These parameters were then globally adjusted to reproduce the results of the following enzymatic reactions: (1) RSK2 phosphorylation by ERK2 started

by the addition of active MKK1 and (2) ppERK2 dephosphorylation by MKP3 (Fig. 6b). The model qualitatively reproduces the basic findings of the in vitro experiments: phosphorylation and dephosphorylation are both repressed in the presence of ORF45 since it competes with both kinases and phosphatases. The refined binding and catalytic parameters were then used as the starting parameter set for fitting the in-cell phosphoERK and phosphoRSK signal triggered by EGF

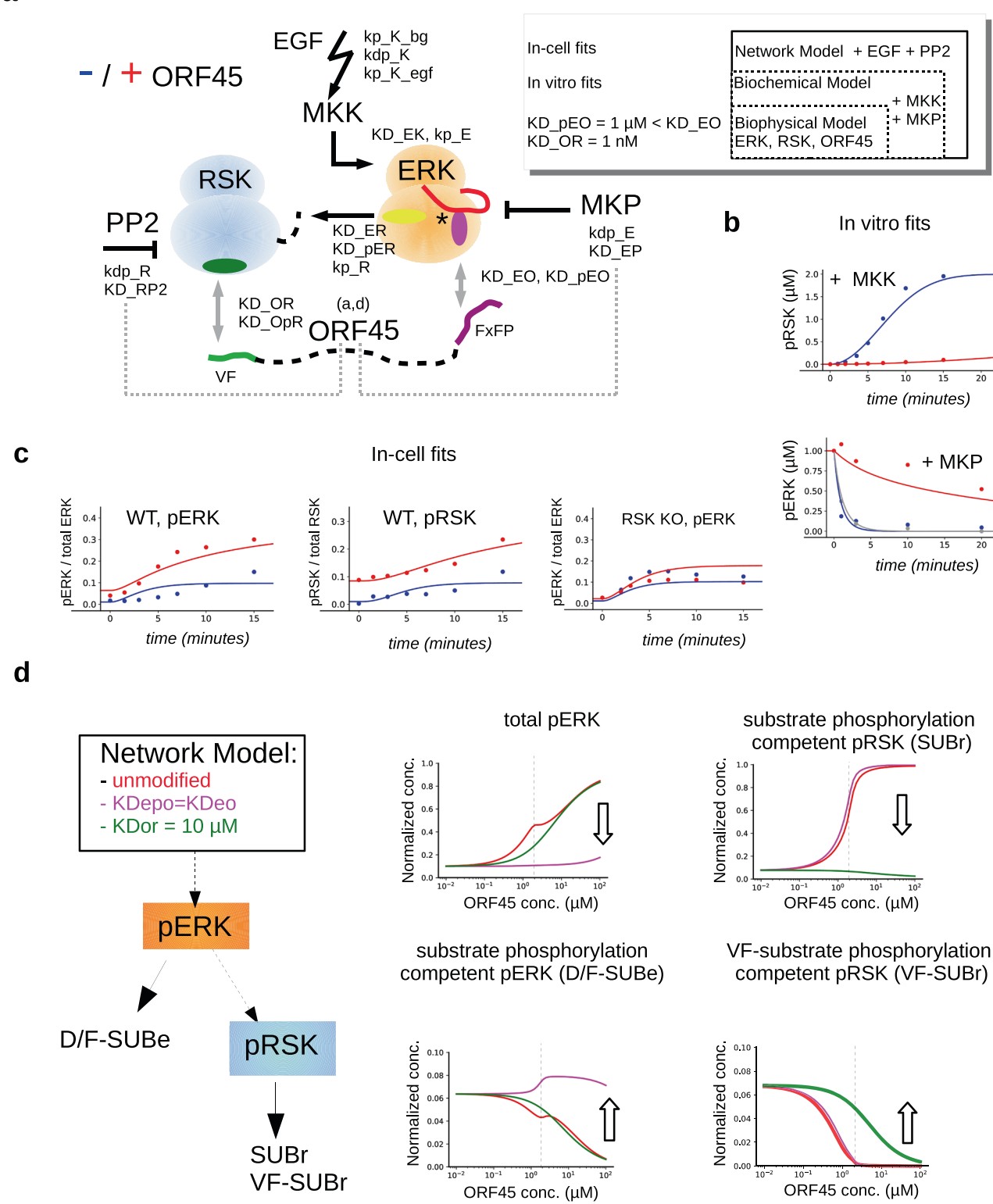

(Supplementary Methods and Supplementary Table 3). In addition, a new phosphatase acting on RSK was introduced and the in-cell protein concentrations were set to realistic values (Network Model). The maximum level of ppERK upon EGF treatment was determined experimentally (~15% of the total ERK level at 15 min; Supplementary Fig. 6). The Network Model — which focuses on the submodule limited to kinase/phosphatase/ substrate binding and catalytic reactions—produces kinetic trajectories that are reasonably similar to the experimental observations regarding ORF45 mediated up-regulation of ERK and RSK phosphorylation in cells (Fig. 6c). The model suggests that the viral protein, instead of being a direct kinase activator, "activates" by protecting both kinases from phosphatases in the cell[7]. Thus, we show that the same basic mechanism using largely the same parameter values can account for the apparent discrepancy between the in vitro and in-cell contexts. ORF45

**Fig. 6 Modeling the effects of ORF45 on ERK and RSK phosphorylation. a** Schematic of the MKK-ERK-RSK Network Model involving phosphatases and ORF45. The inset shows how the Biophysical Model was extended towards Network Model in steps. The full list of reaction parameters, directly measured or introduced into the model based on in vitro or in-cell fits, is shown in Supplementary Table 3. Dashed gray lines indicate that ORF45 binding to RSK or ERK competes with the binding of their respective phosphatases (PP2 or MKP). **b** Results of the parameter fit to in vitro experiments in the absence (in blue) and presence of ORF45 (in red). Panels show the impact of ORF45 on the MKK1-ERK2-RSK2 reaction (+MKK, see Fig. 5a) or on the dephosphorylation of ppERK2 by the MKP3 phosphatase (+MKP, see Fig. 5c; blue: pERK2 + ERK2, gray: ppERK2 + RSK2 + MKP3; red: ppERK2 + RSK2 + ORF45). **c** Results of the global parameter fit to the cell-based experiments. The initial phase (up to 20 min; unaffected by transcription level effects) of the EGF mediated ERK pathway response in HEK293T or in HEK293T-ΔRSK1/2 cells was used to fit the Network Model (see Fig. 1; blue: ORF45 is not expressed; red: ORF45 expression is turned on). In-cell ERK and RSK levels were set to 0.7 and 2 μM, respectively, and plots show the ratio of phosphorylated kinase versus total amount. The level of ppERK at 15 minutes after EGF treatment in WT cells was estimated to be ~15% (see Supplementary Fig. 6). **d** Simulation results with intact (red) and modified Network Models (magenta or green). D/F-SUBe: ERK substrates dependent on D- or F-motif binding, shown with a yellow or magenta ellipsoid on Panel **a**, respectively; SUBr: RSK substrates independent of docking; VF-SUBr: RSK substrates dependent on VF-motif binding and thus on access to the RSK VF-groove, shown with a green ellipsoid on Panel **a**. The in-cell concentration of ORF45 was systematically changed between 0.01 and 100 μM and the steady-state level of activated kinases were calculated upon pathway activation (pERK: phosphorylated active ERK normalized to total in-cell ERK concentration). Furthermore, the fraction of kinase competent for substrate phosphorylation (normalized to the total kinase concentration) is also plotted. The dashed vertical line on the graphs indicates the in-cell concentration of ORF45 used in the original Network Model (1 μM). Arrows indicate how the artificial parameter change affects the phosphorylation levels compared to the intact model.

affects ERK and RSK phosphorylation in opposite ways by protecting them from kinases and from phosphatases. If both kinases and phosphatases are present, as in the in-cell context, it depends on the parameters whether the net effect of ORF45 will be activation or inhibition

Since the Network Model was built after an extensive characterization of critical pathway components inherited from the Biophysical Model, we could use it to investigate how key mechanistic tenets of the model (e.g., the increased binding affinity of pERK (pE) to ORF45 (O), KD_EO > KD_pER, or the nanomolar binding affinity between RSK and ORF45, KD_OR ~ 1 nM) would affect the overall outcome (Fig. 6d). In addition, we could also explore through simulations how different amounts of ORF45 would affect the phosphorylation of different types of ERK or RSK substrates upon pathway activation. At higher ORF45 concentrations the fraction of RSK capable to phosphorylate its classical substrates (SUBr) is increased but RSK docking dependent substrate phosphorylation (VF-SUBr) drops. The fraction of substrate phosphorylation competent ERK is decreased at higher ORF45 levels despite that total pERK increases, because the substrate-binding site of the kinase is occluded by the FxFP motif from ORF45. This quantitative analysis suggested that (1) the increased binding affinity of pERK (pE) to ORF45 (O) is indeed key to the simulated pERK related response (compare curves in red and magenta on the pERK simulation panels on Fig. 6d) and (2) nanomolar binding affinity between RSK and ORF45 is indeed required for efficient RSK SUBr phosphorylation and to limit VF-SUBr phosphorylation by pRSK (compare curves in red and green on the pRSK simulation panels on Fig. 6d). In brief, these simulation results suggest an unexpected signaling role for ORF45: the viral protein modifies the PPI network and channels pERK activity selectively towards one of its substrates, RSK, and ultimately towards a specific class of RSK substrates (SUBr), since VF-SUBr phosphorylation is blocked. Inspired by this finding, we posited that there may be bone fide RSK substrates in the host that depend on NTK docking.

**ORF45-RSK binding reveals a new AGC kinase docking system.** The NTK/VF-motif crystal structure showed how the valine and phenylalanine side-chains snugly fit into a small hydrophobic pocket and highlighted the importance of other neighboring D/E residues involved in electrostatic interactions (Fig. 7a). The structure of the C-lobe by the αG-αH loop region is similar in other AGC kinases (e.g., p70S6K, MSK,

PKB, PKC, ROCK), with notable evolutionarily conserved differences: the electrostatic properties of this kinase domain region differ greatly among AGC kinase members. The RSK surface is the most positive but in PDK1, the most distant RSK AGC kinase relative, this surface is negative, while in MSK1, the closest relative, is neutral. We observed that VF-motif containing proteins ORF45 or BMF (see later) bound to RSK2 but not to MSK1, suggesting that VF-motif groove binding is AGC kinase specific.

We postulated that the viral protein may tap into a hitherto unknown linear motif-based docking system in the host, similarly to the FxFP-motif that uses the F-site on ERK. First, we addressed if VF-type motifs may exist in viral or other pathogenic bacterial proteins by carrying out a motif search with sequence patterns where the VF sequence was flanked by at least one aspartate/glutamate residue at either or both sides using SliMSearch (e.g., D/ExVF or VFxD/E, where x could be any amino acid)[33]. This analysis identified intracellular proteins from pathogens that had been previously known to affect MAPK signaling; for example, the YopM protein from the plague bacterium (*Yersinia pestis*) and the Leader protein from Encephalomyocarditis virus[34,35]. We also searched the disordered part of the human proteome and found a putative RSK binding site in the N-terminal region of PPM1D (8-GVSVFSDQ-15) and in the protein phosphatase A (PP2A) catalytic subunit (1-MDEKVFT-7). Interestingly, RSKs were reported to be dephosphorylated by ILKAP, PPM1D/PP2Cδ/WIP1 or PP2A phosphatases[36–38]. Moreover, we identified dozens of other proteins including some known RSK interacting proteins (e.g., NHE1, FGFRs, and SOS1)[39–41]. Next, we tested the binding of some of these putative VF-motifs to the NTK with MBP pull-down experiments (Fig. 7b, c). The motifs from the bacterial YopM or the viral Leader protein (POLG) as well as the human motifs from Son of Sevenless 1 (SOS1), Bcl2-modifying factor (BMF), and Inactive rhomboid protein 1 (RHDF1) were tested, and all showed comparable RSK2 NTK binding to ORF45 fragments. It is noteworthy that the VF-core region of ORF45 alone (see ORF45_c on Fig. 7c) mediates only weak (high micromolar) binding and the binding affinity increases if this region is N-terminally extended, ultimately reaching strong, low nanomolar binding affinity. This was also the case with the YopM motif, where an 8 amino acid N-terminal extension greatly increased binding. These findings, in combination with the observation that the ORF45(16-76)-F66A mutant did not show detectable binding (see Fig. 7b, Lane 2), suggest that the

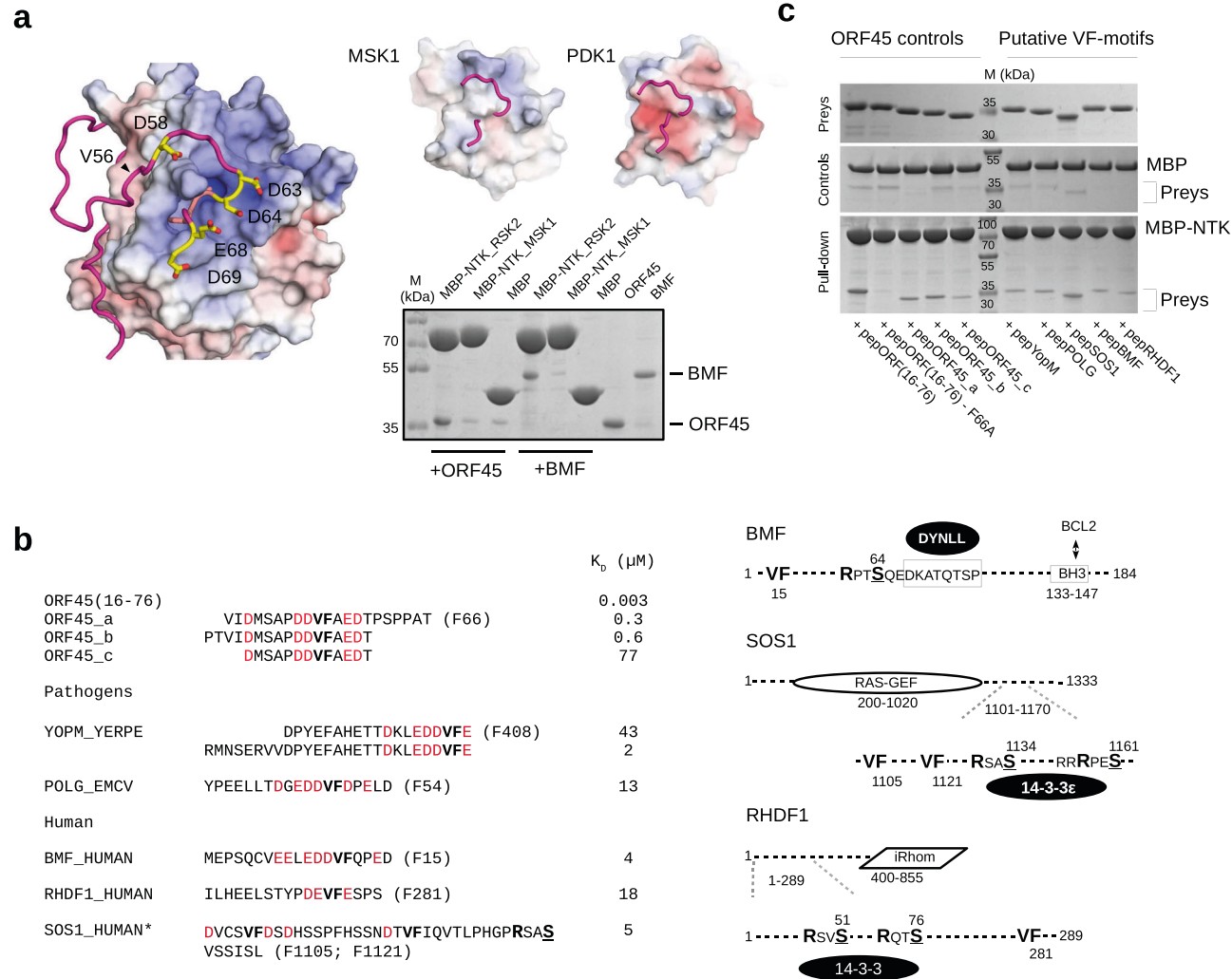

**Fig. 7 VF-motifs from RSK signaling client proteins. a** ORF45 VF-motif (in magenta) docks into a positively charged (blue) AGC kinase domain region via aspartates/glutamates (in yellow). The same surface area for a close RSK relative, MSK1 (PDB ID: 1VZO)[82], and for the most distant AGC kinase relative, PDK1 (PDB ID: 4RRV)[83], is also shown with the ORF45(56-69) region overlaid on the smaller panels at the right. (Blue: +10 eV; Red: −10 eV). The panel below shows the result of an MBP-pull down experiment where binding of GST-ORF45(16-76) and -BMF was tested with MBP-NTK_RSK2, -NTK_MSK1, and MBP as the control (N = 1). The gel was stained with Coomassie protein dye. MBP: maltose binding protein. **b** Selected peptides with VF-motif-like sequences in disordered protein regions. The position of F in the full-length protein sequence (from Uniprot) is shown in brackets. Indicated protein fragments were produced as GST-fusions, purified and their binding affinity to the RSK2 NTK was measured by SPR. The panel on the left shows the schematic of the VF-motif containing human full-length proteins: disordered protein regions are shown as a dashed line and the relative position of VF-motifs and K/RxxS/T phosphorylation sites are indicated. (In BMF, the dynein light chain binding motif is indicated and boxed, BH3 – BCL2 binding region; SOS1 has a Ras GDP exchange factor domain, RAS-GEF; RHDF1, has an inactive rhomboid protease domain, iRhom.) Phosphorylation sensitive interacting partners are shown with black ellipsoids: DYNLL, dynein light chain binding affects cellular BMF localization; for SOS1, 14-3-3 protein binding affects cellular localization and EGF receptor adapter binding; for RHDF1/2, 14-3-3 binding affects TNFα converting enzyme (TACE) activity and ultimately cell surface receptor shredding. **c** MBP pull-downs with purified GST-fusion constructs containing VF-motif peptides (preys) from ORF45, YopM, POLG, SOS1, BMF, and RHDF1 (see Panel **b**). The gel was stained with Coomassie protein dye. Maltose resin was loaded with MBP-NTK or MBP only, where the latter was used to asses unspecific binding of the preys (N = 1). Source data are provided as Source data file.

VF-core motif is necessary but not sufficient for high-affinity binding.

Next, we addressed phosphorylation of RxxS/T sites in human proteins which are putative (BMF) or known physiological substrates of RSK (SOS1, RHDF1)[42–45]. Full-length (BMF) or VF-motif containing longer constructs (SOS1: 1101-1170; RHDF1: 1-289) were expressed, purified, and used as substrates with activated RSK2 in kinase assays (Fig. 8a). These experiments clearly indicated an important role for VF-motif-based docking in RSK mediated phosphorylation: phosphorylation of all constructs diminished in the presence of ORF45, but the phosphorylation of the VF-motif independent EIF4B RxxS/T site remained unchanged.

ORF45 enables more efficient transcription of the KSHV DNA (e.g., by increasing c-FOS phosphorylation) and increased translation of viral proteins in the host (e.g., by phosphorylating EIF4B)[28,46]. Moreover, changes in the human phosphoproteome upon viral infection highlighted several RSK target proteins in addition to c-FOS and EIF4B, such as 40S ribosomal protein S6, Tuberin/TSC2, or GSK3[47]. These known RSK substrates do not contain VF motifs and their phosphorylation is independent of RSK docking. How would ORF45 affect the phosphorylation of VF-motif dependent and independent substrates in the cell? In order to address this question, the computational model was extended and used to calculate RSK substrate phosphorylation

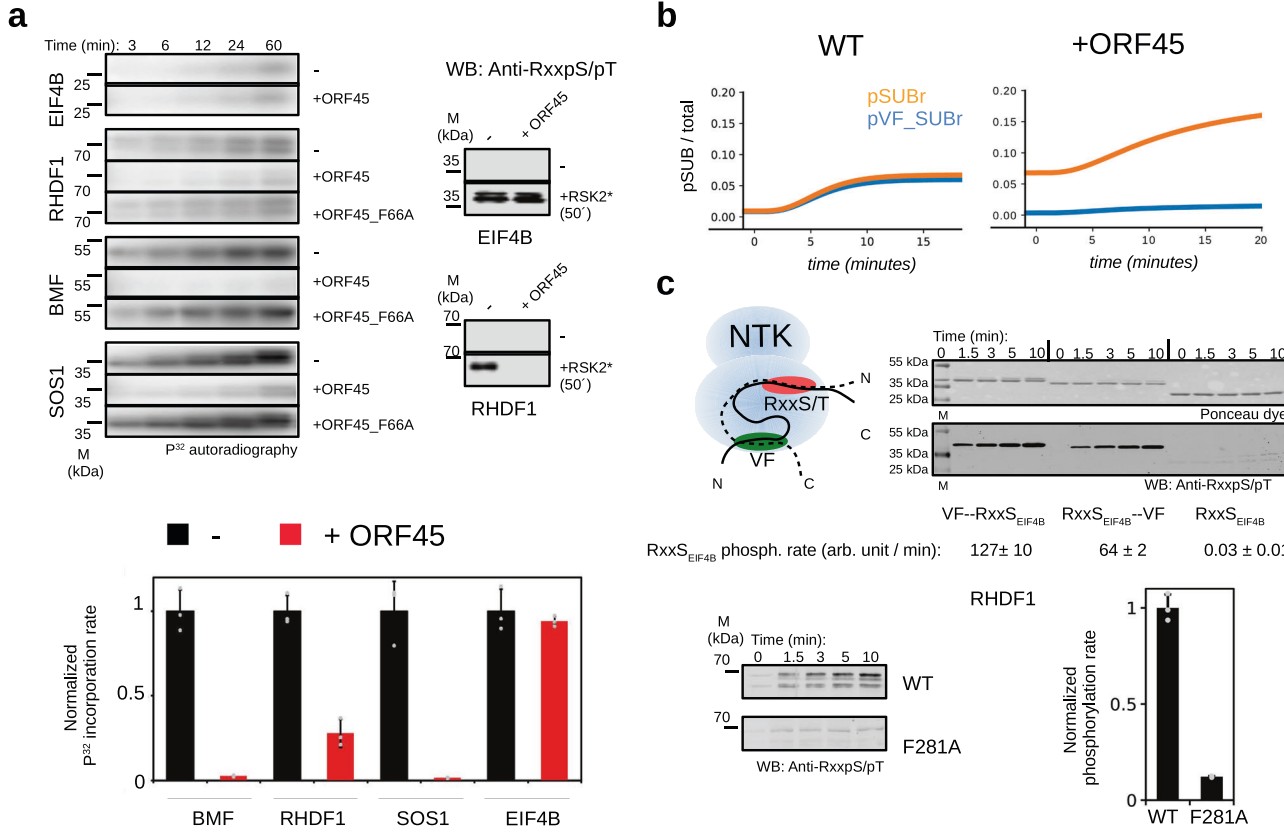

**Fig. 8 VF-motif mediated phosphorylation of RSK substrates. a** In vitro kinase assay results with VF-motif containing human proteins: BMF (full-length), SOS1(1101-1170), RHDF1 (1-289), and with a VF-motif lacking substrate (EIF4B). The panels below show the results of in vitro kinase assays: P$^{32}$-autoradiography based detection by phosphorimaging is shown on the left and the Western-blots using the anti-RxxpS/pT phosphorylation sensitive antibody on the right confirms AGC kinase-specific site (RxxS/T) phosphorylation for EIF4B and RHDF1 after 50 min. 0.5 μM activated RSK2 was mixed with 10 μM GST-fusion constructs (EIF4B: 411-ETQERERSRTGSESSQTGTS-430; full-length BMF, SOS1(1101-1171), and RHDF1(1-289)), and the substrate phosphorylation was monitored in the absence (-) or presence of 2 μM competitor: GST-ORF45(16-76) or the VF-motif mutated version of the latter (ORF45_F66A). The bar charts below show the initial phosphorylation rate normalized to the reaction containing no competitor. Error bars show SD calculated based on three independent experiments. *P*-values of two-sided paired t-tests were the following: BMF, $p = 1.8E-4$; RHDF1, $p = 5.4E-4$; SOS1; $p = 6.4E-4$; EIF4B: $p = 0.5$. **b** Simulation results showing the effect of ORF45 on the phosphorylation levels of VF-motif independent (SUBr) and VF-motif dependent (VF-SUBr) substrates after EGF stimulation. The plots were generated using the Network Model (Fig. 6). **c** Enhanced phosphorylation of the RxxS/T motif through VF-motif-based RSK docking. RxxS/T substrate phosphorylation sites bind in the negatively charged substrate-binding pocket of NTK (depicted with a red ellipsoid; RxxS$_{EIF4B}$ indicates the 411-430 region from EIF4B, see Panel **a**). A VF-motif located N- or C-terminally—VF-RxxS$_{EIF4B}$ or RxxS$_{EIF4B}$-VF, respectively—to this phosphorylation site may enhance the rate of phosphorylation through VF groove (shown with a green ellipsoid) based docking. The phosphorylation rate of 10 μM GST-fusion constructs containing the EIF4B RxxS/T phosphorylation site alone (RxxS$_{EIF4B}$) or in combination with the VF-motif from ORF45 (VF: ORF45_a, see Fig. 7b; VF-RxxS$_{EIF4B}$ or RxxS$_{EIF4B}$-VF; where the F and S from these two motifs were separated by 34 or 20 amino acids, respectively) were examined in kinase assays using 100 nM activated RSK2. The upper panel shows the gel stained with a protein dye (Ponceau) and the lower panel shows the quantitative Western-blot signal obtained with the phosphorylation-specific Anti-RxxpS/pT antibody. Phosphorylation rates (mean) were calculated based on the linear phase of the kinetic plots and the SD was calculated based on three independent experiments (N = 3). The panels at the bottom show the diminished phosphorylation rate of the RxxS motifs in RHDF1 (5 μM RHDF1$_{1-289}$ with 0.25 μM activated RSK2) when the VF motif was mutated (F281A). Source data are provided as Source data file.

levels in cells in the absence and presence of ORF45 (Fig. 8b). These simulations suggested that ORF45 has different impact on VF-motif containing competitive (VF-SUBr) or noncompetitive (SUBr) substrates: the phosphorylation of the latter group increases—because increased RSK phosphorylation directly leads to increased substrate phosphorylation, while the former decreases in the presence of ORF45— because the high-affinity binding site in ORF45 blocks these substrates from accessing the VF-motif binding docking groove on the NTK. These findings suggest that NTK/VF-motif binding could be instrumental in endogenous RSK signaling, and this is hijacked by ORF45. It is noteworthy that ORF45 efficiently blocks RSK mediated phosphorylation of one of the VF-SUBr type signaling clients, SOS1. This protein is particularly interesting in this context as SOS1 is

the GDP exchange factor (GEF) controlling RAS activation: phosphorylation at S1134 and S1161 promotes 14-3-3 protein-dependent removal of SOS1 from the cell membrane and limits ERK-RSK signaling by negative feed-back[41,48]. Intriguingly, the RSK controlled phospho-switch region in SOS1 contains VF motifs optimally located to be able to facilitate the phosphorylation of sites required for 14-3-3 protein binding.

Kinase docking motifs may increase the phosphorylation rate of substrate sites[16]. Therefore, we tested how VF-motif-based RSK docking affects the phosphorylation rate of an RxxS/T type AGC kinase phosphorylation site. We created artificial VF-motif containing phosphorylation reporters where the phosphorylation of the EIF4B RxxS/T substrate-motif was measured in kinase assays (Fig. 8c). We found that the rate of substrate-motif

phosphorylation may be increased by a factor of several thousand if a VF-motif is pasted N- or C-terminally next to the RxxS/T motif. Moreover, we also tested RxxS/T motif phosphorylation in the disordered N-terminal, 14-3-3 binding controlling region of RHDF1. Phosphorylation of these sites was also greatly increased if the VF-motif was intact, albeit the RxxS/T motifs and the VF-motif are separated by more than 200 amino acids. These findings suggest a fairly loose coupling between the RSK-specific VF-motif and the more promiscuous AGC kinase phosphorylation site in terms of relative location and distance. In conclusion, VF-motifs could be instrumental for efficient RxxS/T motif phosphorylation in some of the cognate RSK substrates, and the VF-motif core, depending on the neighboring sequence, may mediate low micromolar binding affinity with RSK signaling clients.

## Discussion

Pathogens may modify the cell's signaling machinery and hijack signaling proteins[3]. ERK2, RSK2, and ORF45 form a stable complex in which the viral protein glues three sequentially acting kinase domains together. ORF45 binding blocks ERK-mediated RSK activation in vitro but it activates RSK signaling in-cells. Here we showed that a signaling network model comprised of the relevant kinase/phosphatase/substrate interactions and enzymatic reactions with realistic parameters can explain these opposing findings (Fig. 6).

Binding between an FxFP-motif from the Elk1 transcription factor and ERK2 had formerly been elucidated by hydrogen/ deuterium exchange mass spectrometry and structural modeling[19,49,50]. The crystal structure of the ppERK2-ORF45(FxFP) complex agrees well with the structural models put forward in these studies, and we posit that the viral FxFP-motif has evolved to bind to activated ERK2 (Fig. 2b). ORF45 binds to RSK2 with low nanomolar affinity but binary binding between these two proteins is functionally relevant only if its impact on ERK is also considered. ORF45 binding to RSK causes longer residency time for ERK in the complex as it lowers the dissociation rate ($k_{off}$) of the ERK-RSK heterodimer. Although this also limits activation of nonphosphorylated ERK by MKK1, because RSK and MKK1 both compete for the same binding site, the net dynamic outcome is that ORF45 protects ppERK from dephosphorylation because phosphatases and RSK also compete for the same binding site. In addition, ORF45 may also protect RSK from its own phosphatases.

The VF-motif binding groove by the αG/αH region at the bottom of the NTK C-lobe, to our knowledge, has so far not been reported as a linear motif binding surface. However, a clinical study found the F268S mutation to be associated with an RSK signaling-related genetic disease (Coffin-Lowry syndrome)[51], suggesting that intact RSK docking is important for normal signaling (Fig. 2d). By analogy to the D- and F-grooves used by different MAPKs, and the PIF-pocket located on the N-lobe of many AGC kinases involved in phosphorylated HM recruitment[52], we speculate that similar docking grooves on the C-lobe of AGC kinases may be important for other family members beyond the NTK of RSK.

The YopM protein from the plague bacterium (*Yersinia pestis*) is an exotoxin that gets injected into host cells to modify immune cell signaling and is also known to bind to and activate RSK. The RSK binding region of YopM was mapped to the disordered C-terminal region of the bacterial protein; DKLEDD**VF**E*, where alanine replacements in the last 6 amino acids completely eliminated YopM-RSK binding and greatly decreased the virulence of those bacterial strains that had these mutations[34,53]. The YopM C-terminus is very similar to the ORF45 VF-motif characterized in our study. Moreover, it was shown that the presence of YopM in the host cell causes increased RSK phosphorylation, without increased ERK phosphorylation, due to selective protection from RSK phosphatases[54]. Based on our studies, we speculate that the phosphatase and YopM may compete for the same docking surface on the NTK. Intriguingly, the herpes viral protein, ORF45, went one step further in manipulating ERK-RSK signaling: it acquired an FxFP-type motif that allowed protection from host phosphatases at the level of the MAPK, too. This is in agreement with the observation that ORF45, in addition to increasing the level of phosphoRSK, also caused increased ERK phosphorylation (Fig. 1a). VF-motifs may be functional in other viruses. For example, the Leader (L) protein from Encephalomyocarditis virus (ECMV_POLG) has a putative VF-motif that appears to be instrumental for increased RSK phosphorylation in infected cells[35].

ORF45 binds RSK with a "perfected" NTK binding sequence that extends beyond the VF-motif core. Compared to the strong, low nanomolar binding affinity of the ORF45(16-76) construct, shorter VF-motif-containing constructs (e.g., ORF45$_{60-70}$ or ORF45$_{56-76}$; see Supplementary Fig. 4c and Fig. 7c) bound weaker. Weaker binding affinity could be more typical between NTK and its bona fide signaling clients. These client proteins often harbor known or putative K/RxxS/T phosphorylation target sites for RSK suggesting that VF-motifs may also have functional relevance in cognate RSK → substrate phosphorylation. RSK2 is known to phosphorylate SOS1, and our analysis suggests that this regulatory phosphorylation—involving 14-3-3 protein binding and affecting the membrane localization of SOS1 in cells—could be facilitated by the newly identified VF-motifs located nearby[41,43]. The cell death controlling Bcl-2 modifying factor (BMF) is known to be regulated by MAPK mediated phosphorylation in its N-terminal phosphoregulatory region[55], and this region has an RSK binding VF-motif as well as a phosphorylation target site possibly affecting the binding of this proapoptotic protein to the dynein light chain (DYNLL)[42]. RHDF1/2, also called iRhom1 and 2, are regulatory proteins affected by ERK-RSK signaling and are known to control the activity of a key cell membrane protease (TACE) affecting cell surface receptor shedding[56]. This adaptor protein is regulated by MAPK and RSK phosphorylation where the latter may be facilitated by its VF-motif[44,45]. These observations suggest that VF-motif-based RSK docking is instrumental in RSK signaling.

Many known RSK substrates have an optimal basophilic target site: RxRxxS/T (R, arginine; S, serine; T, threonine; and x, any amino acid)[57,58]. Phosphorylation of a target motif could be facilitated by VF-motif-based RSK docking, especially in the case of less optimal sites (e.g., K/RxxS/T); similarly as for MAPKs where docking motifs are required to efficient phosphorylation of S/TP sites (P, proline)[31,59]. Moreover, a RSK-specific docking motif next to a basophilic phosphorylation target motif could increase the specificity of phosphorylation by RSK, since many AGC kinases have a common basophilic target site preference (e.g., RSK, p70 S6K, and Akt/PKB all have the same RxRxxS/T optimal consensus target motif). ORF45 shunts ERK activity selectively towards RSK, but in the presence of ORF45 this is only translated into increased downstream phosphorylation for VF-motif independent RSK substrates. Conversely, phosphorylation of VF-motif dependent substrates (e.g., BMF, RHDF1/2, SOS1) would be suppressed because of the competition by the perfected viral motif (see Fig. 8b). Moreover, blocking phosphatase-mediated dephosphorylation and SOS1 dependent negative feedback by ORF45 would naturally lead to increased phosphorylation of VF-motif independent RSK substrates. Speculatively, ORF45 mediated selective manipulation of RSK substrates in the host cells offers advantage to KSHV, for

example, activation of RSK1/2 was shown to promote motile/invasive gene program in epithelial cells[48].

In conclusion, we demonstrated that the KSHV ORF45, a non-catalytic protein, hijacks a ubiquitous cell growth-promoting signaling complex via kinase docking motif-based trickery to alter the MAPK signaling intensity, duration, substrate specificity, and ultimately the outcome of the signaling output. Conversely, the pharmaceutically relevant inhibition of MAPKs—unfortunately with lots of unwanted pleiotropic effects at present—may be approached by similarly "smart" strategies that target MAPK-partner protein complexes[60,61].

## Methods

**Cell culture methods**. For making the ORF45 expressing cell lines, HEK293T (ATCC, #CRL-3216) or HEK293T-ΔRSK1/2 cells were transfected with pEBDTet vectors containing the full-length KSHV ORF45 cDNA[62]. HT-ORF45 (HEK293-Tet-ON) stable cell lines were established by keeping the cells under puromycin (HEK293T) or hygromycin (HEK293T-ΔRSK1/2) for two weeks, then expression of FLAG-tagged ORF45 was monitored by Western-blots after doxycycline (DOX) treatment (2 μg/mL) in DMEM containing 10% FBS. Cells were kept under puromycine (5 μg/mL) or hygromycin (150 μg/mL) selection and were serum starved for 16 h before stimulation with 100 ng/ml EGF.

**Protein expression and purification**. All proteins were expressed in *E. Coli* BL21(DE3) strain or Rosetta(DE3) pLysS using modified pET vectors (Novagen). Human RSK2 (Uniprot ID: P51812) was expressed as a GST-fusion or MBP-fusion protein with an N-terminal GST/MBP and C-terminal hexa-histidine tag and double-affinity purified (on Ni-NTA and glutathione or maltose resin, Qiagen). After cleaving the GST- or MBP-tag off by the tobacco etch virus (TEV) protease, RSK2 was further purified on a ResourceS anion exchange column (GE Healthcare). Human nonphosphorylated ERK2 was co-expressed with lambda-phosphatase[15]. Phosphorylated human ERK2 was produced by co-expressing it with a constitutively active GST-tagged MKK1 construct (MKK1_4D; S218D/M219D/N221D/S22D). ERK2 was produced with an N-terminal hexahistidine tag, subjected to affinity purification using Ni-NTA resin and the tag was cleaved off by the TEV protease. The sample was then further purified on ResourceQ and MonoQ cation exchange columns (GE Healthcare). The phosphorylation state of the purified samples (nonphosphorylated or double-phosphorylated) was confirmed by mass spectrometry. ORF45(16-76) or shorter versions of this were expressed as GST-fusion proteins. The cell lysate was subjected to affinity purification using glutathione resin, the GST-tag was then cleaved off by the TEV protease and the sample was boiled in the presence of 1 mM TCEP for 5 min to precipitate most proteins. The sample was then centrifuged, filtered and purified by reversed-phase HPLC using a ReproSil 300; 5 μm; 250 × 10 mm column. The sample was eluted with a gradient using solvent A (0.1% formic acid in water) and solvent B (0.1% formic acid in acetonitril) and were lyophilized.

The RSK2 N-terminal kinase domain (NTK, 53-359) was expressed as an N-terminal MBP-fusion protein with a C-terminal His6-tag. The MBP was removed by TEV protease and the protein was further purified on a ResourceS anion exchange column (GE Healthcare) and a Superdex 75 gel filtration column (GE Healthcare). For crystallization, the NTK (39-351) construct with an N-terminal hexa-histidine tag was co-expressed with GST-lambda-phosphatase using a bicistronic modified pET vector (MG950), and purified as described above.

Biotinylated RSK2 for the SPR experiments was co-expressed with the BirA ligase in *E. Coli* and purified as described above but this construct retained an N-terminal AviTag after TEV cleavage[63]. Longer ORF45 peptides were produced with a C-terminal cysteine (see earlier) and labeled with 5-(iodoacetamido)fluorescein. Shorter peptides were chemically synthesized with standard Fmoc/tBu SPPS strategy and labeled similarly. VF-motif containing various proteins (Fig. 7b, c) were expressed as GST-fusion proteins with a C-terminal 6xHis-tag in *E.coli*. After double affinity purification, proteins were subjected to ion-exchange chromatography (HiTrap Q, GE Healthcare) and used as analytes to measure their binding affinity to RSK2 NTK by SPR (Fig. 7b).

The RSK phosphorylation probe contained the 411-ETQERERSRTGSESSQTGTS-430 EIF4B region (where S422 is underlined; RxxS_EIF4B). The sequences of the VF-RxxS_EIF4B and the RxxS_EIF4B-VF phosphorylation reporters were the following: VIDMSAPDDVFAEDTPSPPATGGGGSGGGGSGSETQERERSRTGSESSQTGTS or ETQERERSRTGSESSQTGTSGRVIDMSAPDDVFAEDTPSPPAT, respectively. These were expressed as GST-fusion proteins with a C-terminal 6xHis-tag in *E. coli*, and purified by affinity chromatography on Ni-NTA and glutathione resin. The NTK (23-348) of MSK1 (Uniprot ID: O75582) was expressed as N-terminal MBP-fusion construct with a C-terminal 6xHis-tag in *E.coli*, purified by affinity chromatography on Ni-NTA purification and then loaded to maltose resin.

Activated full-length RSK2, used in protein kinase assays, were expressed using the Bac-to-Bac Baculovirus Expression System (ThermoFisher Scientific) with an N-terminal 6xHis-tag. The SF9 cell lysate was subjected to Ni-NTA purification

and the eluted sample was further purified on a 1 ml HiTrap Q-Sepharose (GE Healthcare) ion-exchange chromatography column.

Full-length human MKP3 (Uniprot ID:Q16828) was expressed in *E. coli* as a fusion construct with an N-terminal MBP and C-terminal 6xHis-tag and purified by double-affinity chromatography. Sequences of oligonucleotides used for cloning are listed in Supplementary Table 4.

**Protein crystallization and X-ray structure solution**. The ORF45(27-40) peptide was produced by chemical synthesis using the Fmoc strategy. After the last ion-exchange purification step double-phosphorylated ERK2 (ppERK2) was concentrated to 12 mg/ml and then supplemented with 10% glycerol and stored at −80 °C. The sample was mixed with the ORF45(27-40) peptide in 1:1.5 molar ratio, supplemented with AMPPNP (1 mM) and 2 mM TCEP. Custom crystallization screens with this sample gave crystals in which the F-site was blocked by crystal packing. Parallel to these crystallization trials we determined the crystal structure of ERK2 bound to a D-motif peptide (p28) that was covalently linked to the MAPK and noticed that this complex crystallized in a packing arrangement where the F-site was open. We repeated the same crystallization trials but now also mixing in p28 in stoichiometric amounts. This sample gave very thin and long needles that could be grown thicker by replacing AMPPNP with another ATP competitive ERK inhibitor (ERKi). Crystallization was carried out in hanging drop diffusion setups having 1.25 M NaCl in the reservoir. Crystals finally grew in 0.1 M Tris pH 8.3, 20% PEG 8000. Crystals were flash cooled in liquid nitrogen after supplementing the drop with 25% glycerol as cryoprotectant. Data were collected at PETRA III beam line P14, in Hamburg, and data were processed using XDS[64] and merged and scaled with AIMLESS[65]. The structure of the ppERK2-ORF45(27-40) protein-peptide complex was solved by molecular replacement in PHASER[66] using the ppERK2 crystal structure (PDB ID: 2ERK) as the search model[67]. The asymmetric unit contains one ppERK2-p28-ERKi-ORF45(27-40) complex where 9 amino acids of the 14 residue long ORF45 peptide could be traced in the electron density. The structure was refined using PHENIX[68] (see Supplementary Table 1).

For the RSK2 NTK-ORF45(16-76) binary complex, RSK2 NTK and ORF45(16-76) were mixed in 1:1.5 stoichiometric ratio, gel-filtrated and the complex was concentrated to 10 mg/ml. The sample was supplemented with 2 mM AMPPNP. Crystallization trials were carried out in the hanging drop crystallization setup where 0.75 M NaCl was in the reservoir. Crystals grew in 2–3 days in ~12% PEG 6000, HEPES pH 7.5 and microseeding was necessary to obtain crystals suitable for X-ray diffraction analysis. Crystals were flash-cooled in Parabar 10132 (Hampton Research). Data were collected at EMBL PETRA III beam lines, Hamburg. The crystal structure was solved by molecular replacement using the RSK2 NTK as the search model (PDB ID: 3G51). The crystallographic model has 6 complexes in the asymmetric unit. ORF45 was manually built in Coot[69]. Structure refinement was done using PHENIX with NCS restraints for corresponding polypeptides[68]. Region 23-69 from the ORF45(16-76) peptide could be built in the final crystallographic model, where the C-terminal VF-motif region was very similar in all complexes, but N-terminal ORF45 peptide regions displayed some variation.

**HDX-MS**. H/D Exchange Mass Spectometry (HDX-MS) was performed at the Biomolecular and Proteomics Mass Spectrometry Facility (BPMSF) of the University California San Diego, using a Waters Synapt G2Si system with HDX technology (Waters Corporation) according to methods previously described[70,71]. The binary ERK2-RSK2 complex was subjected HDX-MS analysis and results of this were compared to that of the ERK2-RSK2-ORF45(16-76) sample. Samples were gel-filtrated in 20 mM HEPES pH = 7.5, 150 mM NaCl and 1 mM TCEP. This analysis identified two NTK peptides and one CTK peptide that showed different degrees of protection when ORF45 was present. Samples were analyzed with 3 technical replicates and the deuterium uptake was corrected for back-exchange (see Supplementary Table 2).

Deuterium exchange reactions were performed using a Leap HDX PAL autosampler (Leap Technologies, Carrboro, NC). $D_2O$ buffer was prepared by lyophilizing 20 mM HEPES pH ~7.5, 150 mM NaCl, and 1 mM TCEP initially dissolved in ultrapure water and redissolving the powder in the same volume of 99.96% $D_2O$ (Cambridge Isotope Laboratories, Inc., Andover, MA) immediately before use. Deuterium exchange was measured in triplicate at each time point (0, 0.5, 1, 2, 5 min). For each deuteration time point, 4 μL of protein was held at 25 °C for 5 min before being mixed with 56 μL of $D_2O$ buffer. The deuterium exchange was quenched for 1 min at 1 °C by combining 50 μL of the deuteration reaction with 50 μL of 3 M guanidine hydrochloride pH 2.66. The quenched sample (90 μL) was then injected in a 100 μL sample loop, followed by digestion on an in-line pepsin column (Immobilized Pepsin, Pierce) at 15 °C. The resulting peptides were captured on a BEH C18 Vanguard precolumn, separated by analytical chromatography (Acquity UPLC BEH C18, 1.7 μm 1.0 × 50 mm, Waters Corporation) using a 7–85% acetonitrile gradient in 0.1% formic acid over 7.5 min, and electrosprayed into the Waters Synapt G2Si quadrupole time-of-flight mass spectrometer. The mass spectrometer was set to collect data in the Mobility, ESI + mode; mass acquisition range of 200–2000 (*m/z*); scan time 0.4 s. Continuous lock mass correction was accomplished with infusion of leu-enkephalin (*m/z* = 556.277) every 30 s (mass accuracy of 1 ppm for calibration standard).

For peptide identification, the mass spectrometer was set to collect data in mobility-enhanced data-independent acquisition (MS$^E$), mobility ESI + mode instead. Peptide masses were identified from triplicate analyses and data were analyzed using the ProteinLynx global server (PLGS) version 2.5 (Waters Corporation). Peptide masses were identified using a minimum number of 250 ion counts for low-energy peptides and 50 ion counts for their fragment ions; the peptides also had to be larger than 1500 Da. The following cutoffs were used to filter peptide sequence matches: minimum products per amino acid of 0.2, minimum score of 7, maximum MH + error of 5 ppm, and a retention time RSD of 5%. In addition, the peptides had to be present in two of the three ID runs collected. The peptides identified in PLGS were then analyzed using DynamX 3.0 data analysis software (Waters Corporation). The relative deuterium uptake for each peptide was calculated by comparing the centroids of the mass envelopes of the deuterated samples with the undeuterated controls following previously published methods[72]. For all HDX-MS data, samples were analyzed with 3 technical replicates. Data are represented as mean values ± SEM of three technical replicates. The deuterium uptake was corrected for back-exchange using a global back exchange correction factor (typically ~25%) determined from the average percent exchange measured in disordered termini of various proteins[73]. ANOVA analyses and t tests with a p value cutoff of 0.05 implemented in the program, DECA, were used to determine the significance of differences between HDX data points. Deuterium uptake plots were generated in DECA (github.com/komiveslab/DECA) and the data are fitted with an exponential curve for ease of viewing.

**SAXS analysis and structural modeling.** Proteins were mixed in equal stoichiometric ratio and gel-filtrated on a Superdex 200 Hiload 16/600 (GE Healthcare) column, concentrated and dialyzed in 20 mM Tris, pH 8.0, 150 mM NaCl, 1 mM DTT, 1 mM TCEP, 10% glycerol using small volume dialysis buttons. In order to preclude buffer scattering subtraction problems, equilibrated dialysis buffer was used as the control buffer in the SAXS measurement as well as in the dilution series. Control buffers were always handled parallel to protein samples (e.g., freezing and thawing at the same time), as this was important for best buffer correction for good SAXS measurements. SAXS measurements were performed at the BM29 beamline at ESRF or at the P12 beamline at EMBL-Hamburg (PETRA) (Supplementary Table 5). Data were analyzed using the ATSAS program package[74]. Primary data analysis was performed in PRIMUS. To minimize the inter-particle effect on the scattering curve, a dilution series of concentrated stoichiometric complexes from ~8 to ~0.4 mg/ml were measured. Only minor concentration effect was observed during the measurement which was excluded by manual merging (see Supplementary Fig. 3a).

Rigid body modeling was done with CORAL[72]. ERK2 bound to the C-terminal docking motif of RSK2 (PDB ID: 4NIF), the CTK and the NTK (PDB ID: 3G51) were treated as separate rigid bodies. The linker between the docking motif and the CTK, the interdomain linker between NTK and CTK, the terminal elements, and the relative orientation of the kinase domains were allowed to change freely.

The overall logic in the construction of the ternary complex model was the following: we took the crystal structures of three known binary complexes—ERK2-CTK(RSK1), ORF45-NTK(RSK2), and ORF45-ERK2—and used the SAXS data to guide the quaternary structure refinement of these rigid bodies connected with flexible linkers (since the NTK is connected to the CTK and the C-terminal VF region of ORF45 is linked to its N-terminal FxFP region flexibly). We assumed that the binary contacts for ERK2-CTK, ORF45(VF)-NTK, ORF45(FxFP) binding will be the same as in the ternary complex observed in their binary crystal structures and that the linker between the NTK and CTK of RSK2 links these two globular domains flexibly. Since the binary crystal structures do not have information on all residues because some of the protein regions are likely flexible/disordered, these terminal and linker regions contribute to scattering notwithstanding, we introduced them as flexible regions with dummy atoms and allowed them to move freely. The disordered linkers provided steric restraints on the possible arrangements of the globular binary complexes in SAXS data-driven structural modeling and helped to find the best quaternary arrangement between the three globular kinase domains, selected based on the model with the best $\chi^2$ given by CORAL. The binary contacts within the ternary complex—originally taken from the binary crystal structures—were validated by HDX-MS experiments (Fig. 3a). The real challenge was to find out how the NTK as a globular domain may be incorporated into the ternary complex given that its association with ORF45 is mediated by the C-terminal VF-motif region of the latter but its N-terminal FxFP region binds to ERK2 (which binds to the CTK that is flexibly linked to NTK). The final ternary complex model was consistent with the observation that the NTK's substrate-binding pocket and active site are accessible for substrate binding and phosphorylation (Fig. 5b). More specifically, the ternary ERK2-RSK2-ORF45 model was created by using the crystal structures of the ppERK2-ORF45, ERK2-CTK (PDB ID: 4NIF), and NTK-ORF45(16-76) binary complexes. First, the CTK-ERK-ORF45(FxFP) ternary complex was created by superimposing the ppERK2-ORF45(FxFP) crystal structure with the ERK2-CTK binary complex using HADDOCK[75]. The CTK-ERK-ORF45(29-39; FxFP) and the NTK-ORF45(52-69; VF) sub-complexes were treated as two separate rigid bodies.

CORAL was used to find the best global arrangement to the SAXS data where the intervening region between the FxFP and VF motifs in ORF45 (region 40–51), terminal elements (N- and C-terminal, 1–45 or 732–740, respectively, which were not part of the crystallographic complexes), and the linker (RSK2, 349-415) between the NTK (RSK2, 46-348) and CTK (RSK2, 416-732) were allowed to move free[74]. Finally, the missing residues (40-51) between the N- and C-terminal ORF45 peptide were linked together with MODELLER[76]. The final model had a low discrepancy value ($\chi^2 = 1.12$) (see Fig. 3b). A ternary complex model prepared similarly but using the RSK2(NTK)-ORF45 crystallographic model without any modification—where the N-terminal ORF45 FxFP motif segment bound to NTK but not to the FxFP motif binding docking groove on ERK2—gave a worse fit to the SAXS data ($\chi^2 = 1.56$).

**Protein–protein binding assays.** For the firefly luciferase complementation experiments, two fragments of the luciferase (LucN: 1-399 and LucC: 391-550; Uniprot ID: P08659) were sub-cloned into a modified pcDNA3.1 plasmid (Invitrogen) allowing expression of transgenes with an N-terminal FLAG-tag. HEK293T cells were transfected with LucN and LucC containing plasmids using Lipofectamin 3000 in DMEM. All proteins contained the LUC fragments as C-terminal fusions. Cells were kept in 10% FBS for 20 h after transfection and luciferase activity was measured in 96-well plates (Greiner 657160) at 10 min after the addition of 100 μM D-luciferin in a luminescence plate reader (Cytation 3, BioTek).

Fluorescence polarization-based protein–peptide binding experiments were carried out as earlier described[15]. For surface plasmon resonance (SPR) measurements, biotinylated proteins were captured on a Biacore CAP sensor chip using a Biacore S200 instrument (GE-Healthcare). All measurements were done at room temperature using single-cycle setup with the standard Biacore method for CAP chip including double referencing. The $K_D$ was determined based on RU values corresponding to different concentrations of the analyte. Sensorgrams were fit to a 1:1 binding model using the BiaEvaluation software (GE Healthcare).

For luciferase complementation NanoBiT assays, HEK293T or HT-ORF45 cells were transfected with Lbit and Sbit containing plasmids using Lipofectamin 3000 in DMEM. cDNAs were sub-cloned into Lbit and Sbit expression vectors: ERK2 was expressed as Lbit fusions and RSK2 constructs had Sbit fusion tags; and ORF45 was expressed depending on its interaction partner as Lbit or Sbit fusion. Cells were serum-starved for 20 h and luciferase activity was measured in 96-well plates (Greiner 657160) in a luminescence plate reader (Cytation 3, BioTek) after the addition of 10 μM Coelenterazine h (at 10–15 min).

For MBP pull-downs experiments MBP-NTK or MBP (control) were bound to maltose-resin and 20 μl of this was incubated with GST-VF motif peptides or GST-fusion proteins in 10–20 μM concentration in binding buffer (20 mM Tris pH = 8, 100 mM NaCl, 0.1% IGEPAL, 2 mM TCEP). All proteins were expressed in *E. coli* with C-terminal 6 His-tag and purified on Ni-NTA resin before use. The resin was washed three times with 200 μl binding buffer in batch and samples were loaded to SDS-PAGE.

**Protein kinase assays and Western-blots.** Radioactivity based kinase reactions were carried out in 50 mM HEPES, pH 7.5, 100 mM NaCl, 5 mM MgCl$_2$, 0.05% IGEPAL, 5% glycerol, 2 mM DTT using recombinant expressed and purified proteins in the presence of 250 μM ATP and ~5 μCi of [γ-$^{32}$P]ATP. Reactions were stopped with protein loading sample buffer complemented with 20 mM EDTA, boiled, and then subjected to SDS-PAGE. Gels were dried before phosphorimaging by a Typhoon Trio+ scanner (GE Healthcare).

Western-blot results were analyzed using Odyssey CLx imaging system (Li-Cor) and fluorescently labeled secondary antibodies (IRDye 680 RD goat anti-Rabbit 925-68071, IRDye 800 CW goat anti-Rabbit 926-32211; 1:5000 dilution, or IRDye 680 RD goat anti-Mouse 926-68070; Li-Cor; 1:10,000 dilution) or Western-blots were analyzed on a Fluorchem FC2 camera (Alpha Innotech) using anti-rabbit IgG HRP-linked antibody (Cell Signaling #7074S; 1:10,000 dilution) and anti-mouse IgG (Millipore #401215; 1:10,000 dilution) for ECL detection. Total ERK or phosphoERK levels were monitored by using the following antibodies: anti-p44/42 MAPK (ERK1/2) (L34F12) Mouse mAb (Cell Signaling #4696; 1:3000 dilution; referred to as Anti-panERK or Anti-ERK antibody), anti-phospho-p44/42 MAPK (Thr202/Tyr204) Rabbit Ab (Cell Signaling #9101; 1:3000 dilution). The primary antibodies for tubulin as load control was Mouse Anti-alpha-Tubulin (Sigma #T6199; 1:10,000 dilution), for the FLAG-tag and phosphoRSK were anti-Flag (M2) Mouse mAb (Sigma F1804; 1:10,000 dilution) and anti-phospho-p90RSK (S380) (D3H11) Rabbit mAb (Cell Signaling #11989; 1:2000 dilution), respectively. AGC kinase-specific RxxS/T site phosphorylation was detected by using the Phospho-Akt Substrate (RXXS*/T*) (110B7E) Rabbit mAb (Cell Signaling #9614; 1:2000 dilution) as the primary and the anti-rabbit IRDye 680 RD Ab as the secondary antibody. Despite that, all four protein constructs examined (BMF, SOS1, RHDF1, and EIF4B) contain an RxxS/T motif (Fig. 8a), this primary antibody recognized only the sites in EIF4B and RHDF1. Phosphorylation rates were determined from the slopes of linear fits to phosphorimaging or Li-Cor Western-blot signal.

The in-cell level of phosphorylated ERK after EGF treatment (100 ng/mL) was determined using SuperSep Phos-tag (50 μmol/l), 7.5% gels (Fujifilm Wako) where the phosphorylated forms of ERK migrate differently compared to

nonphosphorylated ERK. HEK293T cell lysates were subjected to Western-blotting and ERK bands were identified using the total ERK primary antibody, bands corresponding to double-phosphorylated ERK were identified by using the phospho-specific ERK antibody as described above. The ratio of band intensities was calculated using the Odyssey CLx imaging system (Li-Cor).

**Computational modeling.** The model describes the phosphorylation and dephosphorylation of ERK and RSK and their interaction with the viral protein ORF45. The model consists of a set of ordinary differential equations (ODEs) and is simulated and analyzed using the Python package "SloppyCell" and custom-written scripts in Python[77,78]. Two versions of the model are considered, one that describes the SPR experiments and one that describes the in vitro and in-cell experiments. Both are available as Systems Biology Markup Language (SBML) files (Supplementary Data 1). These models are described in detail in the Supplementary Methods. Regarding the determination of parameters, SPR, in vitro, and in-cell experiments are different in many respects, therefore we allowed the values of shared parameters to moderately vary where deviations added to the overall cost function used for the optimization (Supplementary Fig. 7). EGF stimulation triggers transcriptional level responses after the first 20 min giving rise to more complex regulation which is beyond the scope of the model. However, the initial phase of the EGF response in HEK293T and HEK293T-ΔRSK1/2 cell lines could be modeled without considering transcriptional level effects.

Key binding parameters were directly measured (7 $K_D$, 2 $k_{off}$) and the rest were determined by fitting them to different sets of experimental data: SPR-based kinetic binding (1 $k_{off}$, a and d parameters; Biophysical Model), in vitro enzymatic (1 $K_D$, 3 $k_{off}$ and 3 $k_{cat}$; Biochemical Model) or cell-based (1 $K_D$, 1 $k_{off}$, 4 $k_{cat}$; Network Model) experiments. All parameters (25 in total: $K_D$, $i_{off}$, or $k_{cat}$: $kp$—phosphorylation rate and $kdp$—dephosphoreylation rate) that were used for or obtained from the fitting procedures are listed in Supplementary Table 3. We proceeded in steps by constraining as many parameters as possible with the simpler SPR and in vitro experiments, and fitting only the remaining ones to the in-cell experiments[79]. The in-cell protein concentrations were the following: 0.7 μM ERK (E), 2 μM RSK (R), 1 μM ORF45 (O), 1 μM ERK phosphatase (P) and 1 μM RSK phosphatase (P2), 1.2 μM MKK (K)[80,81].

In order to keep the model manageable, the process of ERK activation and deactivation was simplified. ERK had only two states (and double-phosphorylation was not considered): an activated phosphorylated (pE) and an inactive nonphoshorylated (E). The EGF triggered response was introduced as a first order activating reaction on MKK. In general, phosphorylated enzymes bind with the same binding affinity as their nonphosphorylated form, but phosphorylated RSK binds ERK five-fold weaker[2] ($K_{D\_ER} < K_{D\_EpR}$), and ORF45 binds to phosphorylated ERK ~100 fold stronger than to nonphosphorylated ERK ($K_{D\_pEO} \ll K_{D\_EO}$). The phosphatase acting on RSK (i.e. P2) binds to the VF-motif binding region on the NTK of RSK and thus competes with ORF45 binding. The ERK phosphatase (P) cannot dephosphorylate ERK if the latter is bound to the FxFP motif of ORF45—since access to the ERK activation loop is sterically blocked. The D-SUBe and F-SUBe phosphorylation competent pERK fractions were defined as [pE] and [pE + pER + pER_OR + pEpR_OpR], respectively. SUBr and VF-SUBr phosphorylation competent pRSK fractions were [pRtot] and [pR + EpR + pEpR + EO_EpR + pEO_pEpR], respectively (see Supplementary Methods). The two different types of RSK substrates (VF-SUBr and SUBr) were introduced into the final computational model at low concentrations (at 0.01 μM each). The $K_D$ for RSK-SUBr or -VF-SUBr binding and the catalytic parameter for RSK mediated phosphorylation of these were set to 1 μM and 1 min$^{-1}$, respectively. Introducing the substrates in higher concentrations (at 1 μM each) gave qualitatively similar results.

**Reporting summary**. Further information on research design is available in the Nature Research Reporting Summary linked to this article.

## Data availability
The HDX-MS data was deposited in a ProteomeXchange repository with the accession code: PXD030612. The crystal structure of the ppERK2-ORF45(27-40) and the RSK2 NTK-ORF45 complexes were deposited in the Protein Data Bank with accession codes 7OPM and 7OPO, respectively. The following X-ray structures are available from the PDB: 4NIF, 2ERK, 3G51, and 4NW6. Source data are provided with this paper.

## Code availability
Text file with computer-readable model implementation written in SMBL is provided as a supplementary zip file (Supplementary Data 1: model_SPR.xml, model_invitro-incell.xml, model_with substrates.xml). The models and all code to reproduce the simulations contained in this study can be found at https://github.com/fridoling/ORF45-simulations.

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

## Acknowledgements

This work was supported by the National Research Development and Innovation Office (NKFIH) grants (NN 114309, KKP 126963 awarded to A.R.), VEKOP-2.3.3-15-2016-00011, and by the Hungarian Academy of Sciences (KEP-10/2019). This study was supported by iNEXT (Project 4788). The work was also supported by NIH grants R01DE016680, R01DE026101 to F.Z., by NIH grants R35 GM130389, T32 GM007752 to S.S.T., and by the Italian Association for Cancer Research (Grant AIRC-IG 21556 to A.C.). The HDXMS facility at UCSD is supported by grants "Biomolecular and

Proteomics Mass Spectrometry Facility of UCSD" S10 OD016234 and S10 OD021724. The authors are thankful for the staff at BM29 beamline at ESRF and at the P12 beamline at EMBL-Hamburg (PETRA) for their help in SAXS data collection. All X-ray diffraction data were collected at the EMBL beamline P14 at PETRA.

## Author contributions

A.A. designed and carried out the experiments, analyzed data, and wrote the paper. A.R. supervised research, analyzed data, and wrote the paper. F.G. and A.C. created the computational model and performed simulations. P.S. carried out the SAXS and crystallographic structural work. K.A. measured protein–protein binding by SPR. G.G. contributed to SAXS and HDX-MS data analysis. E.K. and S.S.T. carried out HDX-MS experiments, analyzed the data, and proofed the manuscript. Á.L.P. prepared synthetic peptides and carried out some of the protein-peptide binding assays. I.B. collected X-ray data. F.Z. and E.K. initiated the project and contributed reagents.

## Funding

## Competing interests

The authors declare no competing interests.
