## [Peer Review File · Nature Communications]

REVIEWER COMMENTS

Reviewer #1 (Remarks to the Author):

The key findings of this very interesting and very comprehensive manuscript are the molecular structures of a KSHV (Kaposi Sarcoma-associated Herpesvirus) ORF45 peptide in complex with phosphoERK2 and with RSK. As a result of these two structures and of a set of biochemical data, the authors conclude that the N-terminal domain of the ORF45 protein can simultaneously interact with ERK2 and RSK and thus stabilizes the ERK2/RSK complex and protects it from cellular phosphatases. To achieve this simultaneous interaction with ERK2 and RSK, the N-terminal domain of ORF45 uses a FIFP motif to bind to ERK2, and a VF motif in the context of flanking residues to interact with RSK. Furthermore, the authors provide evidence that the ORF45 VF motif binds into a hydrophobic pocket in RSK and that similar VF containing motifs occur in other cellular, bacterial and viral proteins that have previously been shown to activate RSK. Thus, the way KSHV ORF45 has evolved to activate the ERK2/RSK complex may reflect a conserved functional mechanism that is, perhaps in a varied manner, also used by other pathogens or cellular proteins. This result is therefore very innovative and of broader relevance to the biomedical community.

The reported experiments seem to support the conclusions drawn, although I have to point out that I am not a structural biologist and cannot comment on the details of the structural work presented in this manuscript.

My main criticism relates to the fact that, with one exception, the authors only use *in vitro* biochemical assays and models to study/model/predict the functional relevance of their model, i.e. the role of ORF45 in promoting the formation and stability of a trimeric ORF45/ERK2/RSK complex. Unfortunately, in these *in vitro* assays they can only study an inhibitory effect of ORF45 peptides on the activity of ERK or RSK. I therefore have two major recommendations:

(i) It would be important to study the impact of key mutations in the the ORF45 FIFP motif (mediates binding to phosphoERK2) and VF motif (mediates binding to RSK) on the EGF-induced activation of ERK2 and RSK in cells, using experiments similar to the one shown in Figure 1. The authors would have to generate HEK293 cell lines, in which the expression of ORF45wt (as in figure 1), or mutated ORF45, can be induced by doxycycline.

(ii) To assess the importance of the ORF45-mediated stabilization of an ERK2/RSK complex in the viral life cycle, the authors should introduce mutations in the ORF45 FIFP and VF motifs into a recombinant KSHV genome cloned in a bacterial artificial chromosome and then investigate the ability of the ORF45 mutants to enter the lytic replication cycle.

Minor points:

(i) Fig. 1A,B: the legend mentions HEK293T and HEK293T-delRSK1/2 cells. According to the main text, these cell lines were 'HT-ORF45' cells expressing inducible ORF45, presumably with and without RSK1/2.

(ii) Line 329: Figure 7B should be Figure 7C, or the order of panels in Figure 7 needs to be changed.

(iii) line 341: Figure 7C should be Figure 7B (or change the order of panels in Figure 7).

Reviewer #2 (Remarks to the Author):

The manuscript by Alexa et al describes the biochemical and biophysical characterisation of a tertiary complex formed between a herpesviral protein (ORF45) and two host proteins (the kinases ERK and RSK). The paper describes the authors structural characterisation of ORF45-ERK2 and ORF45-RSK2 structures. The authors further characterised these systems using HDX-MS, SPR, and devised extensive models describing the tertiary complex formation. While the modelling in this manuscript is beyond my expertise, my review focuses primarily on the details of the structural data (X-ray, and HDX-MS).

Overall, this is work that appears to be done at a high standard, and makes an important contribution to understanding how viruses manipulate host kinase signalling. This should be published in Nature Communications. There are however, multiple additional details and supplemental information that will be required to make this manuscript acceptable for publication. Major and minor points are listed below.

1. In figure 2 the authors very clearly show the binding site of RSK with ORF45 in panel C. The Fo-Fc omit map density is clear for the ligand, with excellent agreement with the ORF45 linear motif. For panel B, however, it is very difficult to make out the omit map with the linear motif bound to ERK. Could the authors potentially include a zoom in on the omit map surrounding the linear motif, as this would greatly increase the ability to judge the quality of this fit.

2. To explore the influence of ORF45 on the tertiary complex formation with RSK and ERK the authors utilised HDX-MS. This was combined with SAXS to generate a putative tertiary model. The authors only give a very cursory explanation of the a very small subset of their HDX data (there are three peptides shown in the main figure 3). Much more expansive description of this in the supplemental data would provide validation of the quality of this data set.

There are almost no methods described for this, with it only stating that methods were previously described. The methods simply say there were two NTK peptides with differences, and one CTK peptide with differences. There is no clarification of what the threshold is for significant differences. There is also no discussion on the inclusion of biological or technical replicates. For this data to be acceptable the following HDX information would need to be included.

- a. A full data analysis table as described in the HDX-MS community manuscript (Masson et al Nature Methods 2019). This needs to include % coverage of ERK, RSK, and ORF45, average error, redundancy, etc. Please consults this manuscript for full details.
- b. A more extensive description of the HDX-MS time course, how long proteins were pre-incubated, controls utilised (back exchange, on exchange, fully deuterated controls, etc).
- c. Ideally as source data the authors should include the full set of peptides analysed, and the deuterium incorporation over time with and without ORF45.

Reviewer #3 (Remarks to the Author):

This manuscript presents a wide-ranging and rigorous biophysical-systems biology characterization of the interaction between the KSHV ORF45 protein with ERK and RSK. The authors have worked to characterize these interactions through a variety of methods, and including both protein-protein contact surfaces, enzymatic mechanisms, and more complex dependencies upon function and ternary complex structure, other cellular components, etc. It is an ambitious project and is moving in the direction of biophysical systems biology that is required for ongoing progress in the study of biological systems and their function.

My main concern is that the study is so wide reaching, that it is not clear what the major conclusions are. If I were to title this manuscript, I would have titled it something more clear like "Biochemical and Biophysical Characterization of the impact of HSHV ORF45 on ERK-RSK signaling". I'm not quite sure as to the significance of the current title with respect to the findings, and it is so general that I am not sure it is impactful or meaningful.

My concern may be that I don't quite believe the authors sufficiently demonstrated the concluding sentence of their abstract, which was: "The viral protein exploits a systems level niche in the natural design of protein kinase cascades." The field of systems biology has been around for 20+ years, so this sentence implies certain things will be done and observed. However, I do not see this claim justified by this manuscript.

This may be a difference of fields. I am coming to this as a systems biologist with a solid biophysical background; the authors seem to be biophysicists who are doing solid math modeling and are aware of systems biology concepts. However, in my opinion – the math modeling comes off as technically accurate work that rounds out the paper, but does not advance the narrative in a meaningful manner. Elaboration on that point is as follows:

With regards to the modeling of allostery and the ternary complex (i.e. Figure 4). The discussion seems technically correct on superficial glancing – but I did not dig in deeper as the conclusions seem like non-controversial results that would be discussed in a graduate level class of mathematical modeling of biophysical complexes. I.E. this seems to be a specific example of a well-known general result. If my interpretation is incorrect, this should be better communicated by the authors - they need to better explain how their finding is novel and not a trivial example of a well-known phenomenon.

With regards to the larger model, this also seems to have some issues that would require major revisions and reconsiderations as to presentation.

The larger model is motivated by the desire to reconcile the discrepancies between in vitro and in cell behavior. Scientifically, and with respect to this manuscript, this discrepancy reconciliation is essential. Mathematical modeling is definitely a reasonable approach to ask whether seemingly conflicting outcomes could actually be explained by a single mechanism with variations in abundances between conditions, and/or between rate constants based on truncation mutations, etc. I trust the authors conclusion that it could be.

However, there is major difference between using a model to state that something is possible and proving something. The authors deserve credit for being responsible with their language – they did not claim too much. However, there is so much confusion in the field that it is important for authors to better explain their model and what they can and cannot conclude. It is not uncommon to see someone refer to a demonstrative mathematical model (i.e. one that demonstrates something is not impossible) but misinterpret it to be a rigorous “proof” that such is the behavior of the system.

My suggestion is then to present the model more clearly and to better explain the conclusions that the authors make from it. This would also help the entire narrative of the manuscript, where the "systems" conclusions at times feel tacked on and an attempt to make the paper seem to be more than a comprehensive biophysical characterization.

At minimum, the schematic in Figure 6A needs to be significantly revamped to clearly show how the various protein-protein interactions influence whether or not the various phosphatases can act on their substrate. That is essentially the purpose of the model, but these dependencies are not clearly evident from inspection of the model schematic. This information is buried in the methods, but it needs to be in the schematic. This may help reveal whether the findings of the mathematical work are obvious or if they are non-obvious outcomes that follow from the measured biochemical properties.

The authors should also better communicate the use of the model in the main text. Is this simply saying “this is possible?” or do the authors think their model is right (or mostly right?)? DO the authors interpret this as proof of their conclusions? (Prospective testing of the model is valuable if the authors think their model is right/mostly right.)

It seems like the authors have one set of best fits in their plots of the model "data". They should better explain how they found their best fits (or if these were tuned, “good enough” fits).

The authors mention using the “Sloppy Cell” work of Gutenkunst et al, which provided the modeling framework. However – this presents a striking systems biology conflict. The work of the Sethna lab, as in the Gutenkunst et al work, builds on the concept of “sloppy” models where a large number of very different sets of parameters can result in similar model behaviors that "fit" the data equally well. The concept of “best fit” is hard to explain in this framework. Were the authors looking at many different sets of “sloppy” parameters? Or did they just use the code and deviate from the motivating systems biology of that code? Additional explanation is required on all of these topics.

On quick inspection, the modeling looks technically valid. The authors are clearly very strong biophysically, and the biophysical foundations of modeling seem very well addressed. However, the value of modeling and whether it clearly answered systems questions is not well clear to the reviewer after reading the current version of the current manuscript.

If the model is purely “this is possible” – it may be better to move the modeling figure(s) to supplementary material, but explain it clearly in the main text regarding what it adds, what it cannot add, etc.

Overall - this appears to be a technically excellent body of work that works to understand how a collection of proteins function together through a nice combination of appropriate biophysical methods.

Reviewer #4 (Remarks to the Author):

Re: Nature Communications 21-25149

A viral ORF, ORF45 from herpesvirus KSHV, has binding functions for both the MAP kinase pathway components ERK and RSK and induces persistent phosphorylation of and signaling through both proteins in vivo. The present study presents 1) structural biology that reveal details of docking interactions and overall shape of complexes, and 2) in vitro and in vivo assays of binding. 3) An elaborate kinetic model is offered to attempt explain discrepancies between in vivo results (sustained signaling) and the in vitro results (interaction induced inhibition by ORF45).

The paper builds on prior work from this group on the structure of a complex between ERK2 and the C-terminal kinase domain (CTK) of RSK, and other projects pointing to the existence of interactions and activities among the components described.

Some of the data presented appears quite good, especially the interaction of ORF45 with RSK-NTK (N-terminal kinase domain). The SPR data is also good.

However, the HDX data, SAXS data, and kinetic model are not completely presented, such that it is not possible to comment on the validity of the data or conclusions.

What follows are more detailed comments, both positive and negative.

To answer the specific review questions, to this reader, the highlight of the paper is Figure 2 panels C and D (structure of ORF45 with RSK-NTK), the various SPR binding experiments presented, and assays showing a difference between in vivo and in vitro results on the effects of ORF45 on ERK and RSK phosphorylation. These results are significant. There are flaws in paper.

Main results:

Fig. 1. ORF45 induces ppERK2 and pSer380 of RSK only in the presences of RSK in cells.

Fig. 2. Major discovery of the paper: Fig. 2C ORF45 has an extensive interaction with structure of ORF45 with RSK-NTK encompassing both a conserved dipeptide VF motif as well as the FxFV motif, which is better-appreciated as an ERK2 binding motif. Nice data on the FxFV binding motif in complex with ERK2.

Fig. 3. Some deuterium exchange on the ternary complex (but figure highlighting the NTK of RSK showing protection). HD/X data is poorly described; figures poorly labelled. SAXS data is poorly justified and poorly described. The figure is too small to evaluate. Luciferase complementation over time in cells showing a peak in complex formation in time. Interesting data, but tangential to the rest of the story.

Fig. 4. ppERK2 complex with RSK2-ORF45 is tighter than with ERK2. This is good data showing the complex favors ppERK2.

Fig. 5 A. In vitro, ppERK in complex with ORF45 does not phosphorylate RSK2, whereas ERK2 mixed with an activated MKK1 (MKK1EE) does phosphorylate. The activity is dependent on the F in the VF motif of ORF45. C. The dephosphorylation of pERK is prevented by RSK+OR45.

Fig. 6. An overly complex model is describe in Supplemental Table S2 and Figure S2. Fig. 6 is fits of in vitro and in vivo data to the model, showing how with different model parameters, you can fit both the in vitro inhibition effects and in vivo activation effects.

Fig. 7. Here the paper moves on to informatics and tests on VF-motif containing substrates of RSK. Straightforward and ok, if arbitrary.

Major Problems:

1. However, the story is not logically presented, the data is inadequately described in places, diagrams, especially structures, are inadequately labeled, nomenclature is not used consistently, and structural elements important to the experimental story are not presented in the introduction.

2. A second major issue with the whole paper is the strong allosteric component to the ternary interaction (ORF45-RSK-ERK), which they correctly uncover in Fig. 4C. Apparently, there is something

unique about the ternary complex not learned from the two linear interactions revealed in Fig. 2. Thus, a better paper on this subject would have used the techniques at hand to validate the interactions observed in Fig. 2B, for example by deuterium exchange, and then done more with SAXS analysis to address whether the interactions between RSK and ORF45, unlikely present in the ternary complex, are there or not. Goodness of fits for multiple models?

3. A third issue is that they attempt to explain the discrepancy between in vitro and in vivo data with a very complex and purely mathematical model, presented in the Supplemental Materials. This model contains species that are not likely to occur. The nomenclature is very poor, using 3 letters in place to describe complexes that can have either 3 or two components (and not intuitive). No validation is offered such as confidence intervals or correlation matrices among variables. Then, at the end of the results and into the discussion the reader is expected to buy the conclusions drawn from this inadequately presented model.

Even more specific comments:

1. Explanation of the early part of the story.

a) FxFV, FXFV, and FIFV. Is FIFV an example FxFV, present in ORF45? A better construction is needed at line 126.

b) The idea that the FxFV sequence binds to RSK-NTD was not anticipated. This needs to be discussed in the Results section in the paragraph starting at line 147.

c) The VF-motif is not mentioned in the introduction. First mentioned on Line 152.

2. Figures.

a) Fig. 1 is ok.

b) Fig. 2. Fig. 2A should be better annotated taking into account the multiple references to the VF segment throughout the paper. Fig. 2B, second panel is ok. All of the rest are inadequately labeled. The goal should be that someone with a passing knowledge of protein kinases can understand the view, and what is seen. In general, this problem can be fixed by labeling the secondary structure elements. In the discussion of this figure, panel 2B is not described. There are multiple interactions, and how the ORF45 weaves across the C-terminal domain of the kinase should be described. The fact that the FxFV motif is chaperoned in this 2-protein complex is very interesting and should be discussed, not just mentioned in passing as it is now.

c) Fig. 3a Deuterium is shown is not described. Statistics? Cutoff values? The description in the methods section says for example on line 635 of the methods it says "proteins were mixed" (with no explanation of which proteins). In discussing the SAXS data, something should be said about the

good Kratky plots presented in the Supplemental. There is no discussion of the data collection. There is no acknowledgement to the beam-line or beam line scientist or authorship. No discussion of alternative models and their fitting. Also, it is customary to report on the radius of gyration, the Dmax in addition to Goodness of Fit. There should be a much better discussion of what went into the model, since one of the major goals of this project should be to understand what is special about this ternary complex.

Fig. 4. OK and interesting binding data.

Fig. 5 and 6 See overall comments above.

3. Discussion. The discussion is over-long. The discussion of the model could state "with the number of docking interactions involved and the number of parameters, it is possible to build a model that fits opposing results, as described here in in vitro and in vivo outcomes of ORF45 signaling."

Please find our point-by-point response in normal font after “**Response:**” to the reviewers’ comments (shown in full in *italics*).

Reviewer 1

Reviewer #1 (Remarks to the Author):

The key findings of this very interesting and very comprehensive manuscript are the molecular structures of a KSHV (Kaposi Sarcoma-associated Herpesvirus) ORF45 peptide in complex with phosphoERK2 and with RSK. As a result of these two structures and of a set of biochemical data, the authors conclude that the N-terminal domain of the ORF45 protein can simultaneously interact with ERK2 and RSK and thus stabilizes the ERK2/RSK complex and protects it from cellular phosphatases. To achieve this simultaneous interaction with ERK2 and RSK, the N-terminal domain of ORF45 uses a FIFP motif to bind to ERK2, and a VF motif in the context of flanking residues to interact with RSK. Furthermore, the authors provide evidence that the ORF45 VF motif binds into a hydrophobic pocket in RSK and that similar VF containing motifs occur in other cellular, bacterial and viral proteins that have previously been shown to activate RSK. Thus, the way KSHV ORF45 has evolved to activate the ERK2/RSK complex may reflect a conserved functional mechanism that is, perhaps in a varied manner, also used by other pathogens or cellular proteins. This result is therefore very innovative and of broader relevance to the biomedical community. The reported experiments seem to support the conclusions drawn, although I have to point out that I am not a structural biologist and cannot comment on the details of the structural work presented in this manuscript.

My main criticism relates to the fact that, with one exception, the authors only use in vitro biochemical assays and models to study/model/predict the functional relevance of their model, i.e. the role of ORF45 in promoting the formation and stability of a trimeric ORF45/ERK2/RSK complex. Unfortunately, in these in vitro assays they can only study an inhibitory effect of ORF45 peptides on the activity of ERK or RSK. I therefore have two major recommendations:

(i) It would be important to study the impact of key mutations in the the ORF45 FIFP motif (mediates binding to phosphoERK2) and VF motif (mediates binding to RSK) on the EGF-induced activation of ERK2 and RSK in cells, using experiments similar to the one shown in Figure 1. The authors would have to generate HEK293 cell lines, in which the expression of ORF45wt (as in figure 1), or mutated ORF45, can be induced by doxycycline.

Response: We agree with the reviewer that it is important to examine the impacts of key mutation in the ORF45 FxFP and VF motifs on activation of ERK and RSK in cells. We indeed have examined the impacts of F66A mutation and reported in previous publications. We have shown that F66A mutant is deficient in activation of RSK or ERK in 293T cells (Fu et al., 2015; Avey et al., 2016). To determine the roles of FxFP motif, we have recently generated F32A, F34A, and F32/34A, as well as F32/34/66A mutations in pEGFP-ORF45 plasmid. We transfected these plasmids into 293T cells. Two days after transfection, the cells were serum starved overnight and then treated with TPA (phorbol 12-myristate 13-acetate) or FBS (fetal bovine serum) for 30 minutes. The cell lysates were prepared and analyzed by western blot. Our preliminary data as shown in the figure below suggested that both FxFP and VF motifs are critical for RSK and ERK activation by KSHV ORF45 in cells.

(ii) To assess the importance of the ORF45-mediated stabilization of an ERK2/RSK complex in the viral life cycle, the authors should introduce mutations in the ORF45 FIFP and VF motifs into a recombinant KSHV genome cloned in a bacterial artificial chromosome and then investigate the ability of the ORF45 mutants to enter the lytic replication cycle.

Response: We generated a F66A mutant virus using BAC16 and characterized the impacts of F66A mutations during KSHV lytic replication and primary infection (Fu et al., 2015; Avey et al., 2016). We have shown that the ORF45-F66A mutant failed to cause sustained ERK and RSK activation during lytic reactivation, resulting in dramatic changes of the phosphoproteomic profile of the infected cells. We also noticed a decreased lytic gene expression and much reduced yield of progeny virion of the ORF45-F66A mutant virus in comparison to the wild type or the revertant. These results allowed us to conclude that ORF45-mediated RSK activation plays a critical role in KSHV lytic replication.

To determine the roles of FxFP motif, we have additionally generated F32/34A and F32/34/66A mutants and performed some preliminary experiments. Our results suggested that either of these mutations nearly abolished activation of ERK or RSK during KSHV lytic reactivation (see figure below). We are in the process of further characterizing viral gene expression and progeny virion production.

We have revised the Introduction section accordingly to highlight this line of our previous works (see Line 47): “The N-terminal ORF45 region (~80 aa) has been shown to be required and sufficient for ERK and RSK activation. In particular, F66 located in a short relatively conserved region, referred to as VF motif henceforth, has been identified as the most critical for binding to and activation of RSK. Accordingly, an ORF45-F66A mutant virus failed to cause sustained ERK and

RSK activation during lytic reactivation and had produced less infectious progeny viruses in comparison to the wild type or the revertant^{8,9}”.

Our work described in the current manuscript concentrates on the biochemical/structural investigation of ORF45-ERK-RSK interactions. We would like to keep the focus on the mechanistic basis of VF and FxFP motif mediated binding. Although our unpublished and preliminary data suggested that both motifs are critically important for the activation of ERK and RSK in cells and likely important for KSHV lytic replication and virion production, we would refrain from the dissemination of these results altogether because we feel the current manuscript is fairly broad in its scope. We are planning to present our more detailed and systemic studies on the biological roles of the VF and FxFP motifs in separate publications. We have noticed that these motifs are conserved in some but not in all ORF45 homologues in gammaherpesviruses. We believe that sequences located around and in-between the core binding motifs influence ERK/RSK signaling and have a substantial impact on viral lytic replication. The core motif pattern and structurally less-defined - but functionally important - “fuzzy” interactions made by adjacent and intervening disordered regions collectively affect the impact of ORF45 orthologues on ERK/RSK signaling in the host. This biological system provides a great paradigm to illuminate the biophysical principles governing linear binding motif-mediated signaling complex evolution.

Minor points:

(i) Fig. 1A,B: the legend mentions HEK293T and HEK293T-delRSK1/2 cells. According to the main text, these cell lines were 'HT-ORF45' cells expressing inducible ORF45, presumably with and without RSK1/2.

(ii) Line 329: Figure 7B should be Figure 7C, or the order of panels in Figure 7 needs to be changed.

(iii) line 341: Figure 7C should be Figure 7B (or change the order of panels in Figure 7).

Response:

(i) The fact that these ORF45 expression could be induced in these cell lines is indicated now by inserting 'HT-ORF45' into the figure legend.

(ii-iii) The order of B and C panels in Figure 7 is fixed.

Reviewer 2

Reviewer #2 (Remarks to the Author):

The manuscript by Alexa et al describes the biochemical and biophysical characterisation of a tertiary complex formed between a herpesviral protein (ORF45) and two host proteins (the kinases ERK and RSK). The paper describes the authors structural characterisation of ORF45-ERK2 and ORF45-RSK2 structures. The authors further characterised these systems using HDX-MS, SPR, and devised extensive models describing the tertiary complex formation. While the modelling in this manuscript is beyond my expertise, my review focuses primarily on the details of the structural data (X-ray, and HDX-MS).

Overall, this is work that appears to be done at a high standard, and makes an important contribution to understanding how viruses manipulate host kinase signalling. This should be published in Nature Communications. There are however, multiple additional details and supplemental information that will be required to make this manuscript acceptable for publication. Major and minor points are listed below.

1. In figure 2 the authors very clearly show the binding site of RSK with ORF45 in panel C. The Fo-Fc omit map density is clear for the ligand, with excellent agreement with the ORF45 linear motif. For panel B, however, it is very difficult to make out the omit map with the linear motif bound to ERK. Could the authors potentially include a zoom in on the omit map surrounding the linear motif, as this would greatly increase the ability to judge the quality of this fit.

Response: Fig. 2b panel includes a zoomed-in view of the original panel now where the omit map can be better seen.

2. To explore the influence of ORF45 on the tertiary complex formation with RSK and ERK the authors utilised HDX-MS. This was combined with SAXS to generate a putative tertiary model. The authors only give a very cursory explanation of the a very small subset of their HDX data (there are three peptides shown in the main figure 3). Much more expansive description of this in the supplemental data would provide validation of the quality of this data set.

There are almost no methods described for this, with it only stating that methods were previously described. The methods simply say there were two NTK peptides with differences, and one CTK peptide with differences. There is no clarification of what the threshold is for significant differences. There is also no discussion on the inclusion of biological or technical replicates. For this data to be acceptable the following HDX information would need to be included.

a. A full data analysis table as described in the HDX-MS community manuscript (Masson et al Nature Methods 2019). This needs to include % coverage of ERK, RSK, and ORF45, average error, redundancy, etc. Please consults this manuscript for full details.

b. A more extensive description of the HDX-MS time course, how long proteins were pre-incubated, controls utilised (back exchange, on exchange, fully deuterated controls, etc).

c. Ideally as source data the authors should include the full set of peptides analysed, and the deuterium incorporation over time with and without ORF45.

Response: a) We included a full HDX-MS data analysis table including the requested information (see Supplementary Table 2).

b) A more extensive description is provided in the additional sections in the Supplementary Methods (p15 in the Supplementary Information: HDX-MS more detailed description)

c) Source data is also included now related to Supplementary Table 2 with more peptide HDX-MS data.

Reviewer 3

Reviewer #3 (Remarks to the Author):

This manuscript presents a wide-ranging and rigorous biophysical-systems biology characterization of the interaction between the KSHV ORF45 protein with ERK and RSK. The authors have worked to characterize these interactions through a variety of methods, and including both protein-protein contact surfaces, enzymatic mechanisms, and more complex dependencies upon function and ternary complex structure, other cellular components, etc. It is an ambitious project and is moving in the direction of biophysical systems biology that is required for ongoing progress in the study of biological systems and their function.

My main concern is that the study is so wide reaching, that it is not clear what the major conclusions are. If I were to title this manuscript, I would have titled it something more clear like “Biochemical and Biophysical Characterization of the impact of HSHV ORF45 on ERK-RSK signaling”. I’m not quite sure as to the significance of the current title with respect to the findings, and it is so general that I am not sure it is impactful or meaningful.

Response: For this project we indeed felt that we need to move in the direction of biophysical systems biology – where for us the term “systems biology” refers to the fact that we elucidate an individual protein-protein interaction in the context of other relevant interactions, which made us go beyond simply addressing if a given interaction occurs or not. It turned out fairly soon in the project that the classical biochemical approach will not suffice and that we have to elucidate ORF45 binding in the context of many other MAPK and MAPKAPK mediated binding events. This necessitated mathematical modeling.

We originally planned to do classical biochemical experiments to reveal how ORF45 manipulates ERK signaling. In the end this study indeed developed into a wide-reaching story but this scope was required. We would like to keep the title as is since from a biological perspective it is a significant finding that the viral protein interferes with kinase docking, thus the current title suggests the mechanism, too.

My concern may be that I don’t quite believe the authors sufficiently demonstrated the concluding sentence of their abstract, which was: “The viral protein exploits a systems level niche in the natural design of protein kinase cascades.” The field of systems biology has been around for 20+ years, so this sentence implies certain things will be done and observed. However, I do not see this claim justified by this manuscript.

Response: We accept that from a systems biology perspective, since the “systems” term may have different meaning for researchers coming from different backgrounds, we may not claim that “The viral protein exploits a systems level niche in the natural design of protein kinase cascades”. This sentence was removed from the abstract and was replaced by “The viral protein interferes with the natural design of kinase docking systems in the host.”

This may be a difference of fields. I am coming to this as a systems biologist with a solid biophysical background; the authors seem to be biophysicists who are doing solid math modeling and are aware of systems biology concepts. However, in my opinion – the math modeling comes off as technically accurate work that rounds out the paper, but does not advance the narrative in a meaningful manner. Elaboration on that point is as follows:

With regards to the modeling of allostery and the ternary complex (i.e. Figure 4). The discussion seems technically correct on superficial glancing – but I did not dig in deeper as the conclusions seem like non-controversial results that would be discussed in a graduate level class of mathematical modeling of biophysical complexes. I.E. this seems to be a specific example of a well-known general result. If my interpretation is incorrect, this should be better communicated by the authors - they need to better explain how their finding is novel and not a trivial example of a well-known phenomenon.

Response: We agree that the observed allostery in the case of ternary complex is a specific example of a general phenomenon, nevertheless it needs to be introduced to the reader since the experimental binding curves could only be explained if this phenomenon was taken into account in the biophysical model (and in the larger model later). We argue that this allostery needs to be highlighted and discussed since it is not straightforward for the general reader how this may mechanistically happen in the context of a three-molecule-complex held together by long flexible linkers between the involved structured domains.

With regards to the larger model, this also seems to have some issues that would require major revisions and reconsiderations as to presentation.

The larger model is motivated by the desire to reconcile the discrepancies between in vitro and in cell behavior. Scientifically, and with respect to this manuscript, this discrepancy reconciliation is essential. Mathematical modeling is definitely a reasonable approach to ask whether seemingly conflicting outcomes could actually be explained by a single mechanism with variations in abundances between conditions, and/or between rate constants based on truncation mutations, etc. I trust the authors conclusion that it could be.

However, there is major difference between using a model to state that something is possible and proving something. The authors deserve credit for being responsible with their language – they did not claim too much. However, there is so much confusion in the field that it is important for authors to better explain their model and what they can and cannot conclude. It is not uncommon to see someone refer to a demonstrative mathematical model (i.e. one that demonstrates something is not impossible) but misinterpret it to be a rigorous “proof” that such is the behavior of the system.

My suggestion is then to present the model more clearly and to better explain the conclusions that the authors make from it. This would also help the entire narrative of the manuscript, where the "systems" conclusions at times feel tacked on and an attempt to make the paper seem to be more than a comprehensive biophysical characterization.

Response (to the four paragraphs above): We changed the manuscript regarding the description and presentation of the larger model - hoping that in the revised version this important aspect of the work appropriately advances the whole narrative of the story.

For example, we state now that we use different types of models to explain on the one hand the results of biophysical experiments (Biophysical Model; see Fig. 4c and Supplementary Table 3 column “SPR fit”) and on the other hand two larger models (Biochemical Model, see Fig. 6b, and Supplementary Table 3 column “in vitro fit”; and Network Model, see Fig. 6a,c and Supplementary Table 3 column “in-cell fit”). The Biophysical Model uses experimentally measured real parameters and reveals a nontrivial finding: there is tighter binding and lower k_{on} than expected ($d \ll 1$ and $a \ll 1$) in the ternary complex. We then build the more complex Biochemical Model where we add an upstream kinase and a phosphatase and expand the model with enzymatic

reactions as an intermediate step that leads to the final Network Model. This latter is a simplified model of what may happen after EGF stimulation in cells – since it does not mechanistically address what happens between EGFR activation and MKK1 activation for example – but its strength comes from the fact that we incorporate all known kinase/phosphatase/substrate interactions and enzymatic reactions relevant for the ERK-RSK signaling network.

We show that with this simplified but largely realistic model we can explain how ORF45 may act as an activator in cells but it behaves as an inhibitor in in vitro kinase assays. For example, the model is consistent with the in-cell results of the “RSK KO, ppERK” experiment shown on Figure 6C (right panel): the presence of RSK, which is one of the substrates of ERK, is required so that ORF45 could behave as an efficient activator of ERK phosphorylation. This non-trivial experimental finding could be “reproduced” in silico with those parameters that we use in the final Network Model. We would like to stress that we progressed in a step-wise fashion when we built the Network Model and that the behavior of this more complex model remained consistent with the parameter values determined based on a less complex model (see Supplementary Table 3).

Finally, we used the Network Model (built based on a - to our knowledge - correct mechanistic blueprint and with measured or realistic parameters) as an instrument to understand how ORF45 could channel the signal, and to explain why some of the properties discovered in vitro are relevant for that. Thus, in addition to carrying out a 'comprehensive biophysical characterization' of critical pathway components, our point is that since the pathway is complex, it produces behaviors that are not trivial. So the Network Model could be used as sensitivity check (e.g. how removal of key mechanistic tenets of the model would affect the outcome; see Fig. 6d) and to suggest behaviors that can be hardly guessed by simply looking at the whole network (e.g. differential regulation of general RSK substrates versus VF-motif containing substrates; see Fig. 6d and 7e).

At minimum, the schematic in Figure 6A needs to be significantly revamped to clearly show how the various protein-protein interactions influence whether or not the various phosphatases can act on their substrate. That is essentially the purpose of the model, but these dependencies are not clearly evident from inspection of the model schematic. This information is buried in the methods, but it needs to be in the schematic. This may help reveal whether the findings of the mathematical work are obvious or if they are non-obvious outcomes that follow from the measured biochemical properties.

Response: Fig. 6a is modified. The panel now explicitly shows that binding of ORF45 to RSK or ERK competes with the binding of their respective phosphatases (PP2 or MKP). Furthermore, Figure 6 is changed in other aspects, too: 1) to highlight the hierarchical transition from the correct Biophysical and Biochemical Models into the Network Model, and 2) to clarify how the Network model was used to make predictions on different ERK and RSK substrate phosphorylation levels.

The authors should also better communicate the use of the model in the main text. Is this simply saying “this is possible?” or do the authors think their model is right (or mostly right)? DO the authors interpret this as proof of their conclusions? (Prospective testing of the model is valuable if the authors think their model is right/mostly right.)

It seems like the authors have one set of best fits in their plots of the model "data". They should better explain how they found their best fits (or if these were tuned, “good enough” fits).

The authors mention using the “Sloppy Cell” work of Gutenkunst et al, which provided the

modeling framework. However – this presents a striking systems biology conflict. The work of the Sethna lab, as in the Gutenkunst et al work, builds on the concept of “sloppy” models where a large number of very different sets of parameters can result in similar model behaviors that “fit” the data equally well. The concept of “best fit” is hard to explain in this framework. Were the authors looking at many different sets of “sloppy” parameters? Or did they just use the code and deviate from the motivating systems biology of that code? Additional explanation is required on all of these topics.

Response (to the three paragraphs above): Regarding the SloppyCell-framework, we mainly made use of their code because it provides many features suitable for the simulation of biological models, such as structures to represent experimental data and model variants. In addition, it allows for an exhaustive screening of the parameter space in order to identify an ensemble of parameter sets that are consistent with the data (and in particular avoids the problem of being trapped in a local minimum). So “best fit” in our simulations refers to the fit with the minimum cost among the ensemble members.

We do not think that this choice necessarily involves a deep “systems biology conflict”. We understand the insights of the sloppiness perspective to be twofold. One is an insight about biology: many models of biological systems exhibit sloppy parameter spectra, which suggests that these systems perform their function even if some parameters are varied over orders of magnitude. The second insight concerns the methodology of modeling: given the ubiquity of sloppiness in biological models, predictions based on a single best fit of many parameters are potentially misleading. Instead it would be better to make predictions based on the ensemble of parameter sets that are statistically consistent with the data (given the unavoidable uncertainty in experimental data). Our overall modeling approach in this study has been to determine as many parameters as possible by direct experiment, and to fit the remaining parameters successively in a number of SPR and in vitro experiments where each of which independently determines a subset of the parameters of the full model. In this way we are confident that sloppiness does not threaten our downstream results. In the revised version we back this up by providing an ensemble analysis in a new supplementary figure (Supplementary Fig. 7) that shows that the free parameters determined in the successive fits are reasonably constrained by the data.

On quick inspection, the modeling looks technically valid. The authors are clearly very strong biophysically, and the biophysical foundations of modeling seem very well addressed. However, the value of modeling and whether it clearly answered systems questions is not well clear to the reviewer after reading the current version of the current manuscript.

Response: We are hoping that after the rewriting the modeling section of the manuscript and improving the model related figure the reviewer would now agree with us that this is an invaluable part of the story and it was required to understand what we observed in vitro and in cells.

If the model is purely “this is possible” – it may be better to move the modeling figure(s) to supplementary material, but explain it clearly in the main text regarding what it adds, what it cannot add, etc.

Response: We propose a 'matryoshka' of models, trying to be as close to experimental data as possible. They are all approximations, as all models are, in increasing degrees from the Biophysical to the Network Model. Claiming that a model is true, we believe, it would be naive. However, they were the closest to experimental data that we could formulate and, importantly, they allowed us to formulate hypotheses: (1) we identified allosteric interactions, (2) we explained the in vitro vs in-cell paradox, and (3) we provided a hypothesis about the role of ORF45 in channeling the MAPK

pathway that lead us to new experiments/measurements. All in all, we would say that the model is 'right enough', and turned out to be very useful. Therefore, we kept it in the main part of the manuscript.

Overall - this appears to be a technically excellent body of work that works to understand how a collection of proteins function together through a nice combination of appropriate biophysical methods.

Reviewer 4

Reviewer #4 (Remarks to the Author):

Re: *Nature Communications* 21-25149

A viral ORF, ORF45 from herpesvirus KSHV, has binding functions for both the MAP kinase pathway components ERK and RSK and induces persistent phosphorylation of and signaling through both proteins in vivo. The present study presents 1) structural biology that reveal details of docking interactions and overall shape of complexes, and 2) in vitro and in vivo assays of binding. 3) An elaborate kinetic model is offered to attempt explain discrepancies between in vivo results (sustained signaling) and the in vitro results (interaction induced inhibition by ORF45).

The paper builds on prior work from this group on the structure of a complex between ERK2 and the C-terminal kinase domain (CTK) of RSK, and other projects pointing to the existence of interactions and activities among the components described.

Some of the data presented appears quite good, especially the interaction of ORF45 with RSK-NTK (N-terminal kinase domain). The SPR data is also good. However, the HDX data, SAXS data, and kinetic model are not completely presented, such that it is not possible to comment on the validity of the data or conclusions.

What follows are more detailed comments, both positive and negative.

To answer the specific review questions, to this reader, the highlight of the paper is Figure 2 panels C and D (structure of ORF45 with RSK-NTK), the various SPR binding experiments presented, and assays showing a difference between in vivo and in vitro results on the effects of ORF45 on ERK and RSK phosphorylation. These results are significant. There are flaws in paper.

Main results:

Fig. 1. ORF45 induces ppERK2 and pSer380 of RSK only in the presences of RSK in cells.

Fig. 2. Major discovery of the paper: Fig. 2C ORF45 has an extensive interaction with structure of ORF45 with RSK-NTK encompassing both a conserved dipeptide VF motif as well as the FxFV motif, which is better-appreciated as an ERK2 binding motif. Nice data on the FxFV binding motif in complex with ERK2.

Fig. 3. Some deuterium exchange on the ternary complex (but figure highlighting the NTK of RSK showing protection). HD/X data is poorly described; figures poorly labelled. SAXS data is poorly justified and poorly described. The figure is too small to evaluate. Luciferase complementation over time in cells showing a peak in complex formation in time. Interesting data, but tangential to the rest of the story.

Fig. 4. ppERK2 complex with RSK2-ORF45 is tighter than with ERK2. This is good data showing the complex favors ppERK2.

Fig. 5 A. In vitro, ppERK in complex with ORF45 does not phosphorylate RSK2, whereas ERK2 mixed with an activated MKK1 (MKK1EE) does phosphorylate. The activity is dependent on the F

in the VF motif of ORF45. C. The dephosphorylation of pERK is prevented by RSK+OR45.

Fig. 6. An overly complex model is describe in Supplemental Table S2 and Figure S2. Fig. 6 is fits of in vitro and in vivo data to the model, showing how with different model parameters, you can fit both the in vitro inhibition effects and in vivo activation effects.

Fig. 7. Here the paper moves on to informatics and tests on VF-motif containing substrates of RSK. Straightforward and ok, if arbitrary.

Major Problems:

1. However, the story is not logically presented, the data is inadequately described in places, diagrams, especially structures, are inadequately labeled, nomenclature is not used consistently, and structural elements important to the experimental story are not presented in the introduction.

Response: In the revised version we improved the presentation: the kinase structures are better labeled, the FxFP nomenclature is fixed, the VF motif is introduced better, and the SAXS, HDX-MS data are better described (see at the more concrete response in our answers to the more specific comments).

2. A second major issue with the whole paper is the strong allosteric component to the ternary interaction (ORF45-RSK-ERK), which they correctly uncover in Fig. 4C. Apparently, there is something unique about the ternary complex not learned from the two linear interactions revealed in Fig. 2. Thus, a better paper on this subject would have used the techniques at hand to validate the interactions observed in Fig. 2B, for example by deuterium exchange, and then done more with SAXS analysis to address whether the interactions between RSK and ORF45, unlikely present in the ternary complex, are there or not. Goodness of fits for multiple models?

Response: In HDX-MS experiments we detected prominent protection on the NTK by its VF-motif binding slot only. This suggests that the N-terminal ORF45 region is different in the stable closed ternary complex from what we had observed in the NTK-ORF45 crystal structure (see our response to More specific comment 1b, too) and it would rather bind to ERK with its FxFP motif. We built a ternary complex model prepared similarly as described in the manuscript but using the RSK2(NTK)-ORF45 crystallographic model without any modification - where the N-terminal ORF45 FxFP motif segment bound to NTK but not to the FxFP motif binding docking groove on ERK2. This model gave worse fit to the SAXS data ($\chi^2=1.56$ vs $\chi^2=1.12$), indicating that out of the two alternatives: 1) ORF45 binds as it is in the NTK-ORF45 binary crystallographic complex or 2) the VF motif regions is fixed to the NTK but the FxFP region is allowed to bind to ERK2, the SAXS analysis is more consistent with the latter. See Line 624.

3. A third issue is that they attempt to explain the discrepancy between in vitro and in vivo data with a very complex and purely mathematical model, presented in the Supplemental Materials. This model contains species that are not likely to occur. The nomenclature is very poor, using 3 letters in place to describe complexes that can have either 3 or two components (and not intuitive). No validation is offered such as confidence intervals or correlation matrices among variables. Then, at the end of the results and into the discussion the reader is expected to buy the conclusions drawn from this inadequately presented model.

Response: We revised and updated the nomenclature of the different species, binding and enzymatic parameters of the model described in great detail in the Supplementary Methods. We use

a more consistent nomenclature: for example protein names of a given complex are capitalized (ERK: E; RSK: R; ORF45: O) and the phosphorylated form of the protein is indicated by a “p” (e.g. phosphoERK is pE, and phospho RSK is pR). We agree that many of the complex species considered in the model may only form in small amounts, however, these could all form based on the mechanistic rules of the model, and the modeling methodology that we used does not limit us in this respect. In the revised version of the manuscript, we better show that we use the model not simply to reproduce but also to explain the discrepancy between in vitro and in-cell data. The apparent complexity of the model results from the fact that there are three molecular species (ERK, RSK, ORF45) that can each bind to the two others and form ternary complexes. On top of that, two of these species (ERK, RSK) can occur in phosphorylated and non-phosphorylated forms. This leads to a large number of possible complexes that are all biochemically possible and in turn to a large number of reactions that have to be included in the model. Note, however, that many of these reactions are guided by the same parameters, as can be seen in Section 5 of the modeling part of the Supplementary Methods. For this reason, the model complexity, which is typically understood in terms of the number of free parameters, is actually not as high as it seems. We do not exactly know what the reviewer means by “purely mathematical”, but we have taken great efforts to build our model on mechanistic grounds and to determine parameter values based on experimental data as much as possible. Confidence intervals and correlation matrices for variables are meaningful for the validation of statistical models, which describe the relationships between different quantities measured in the same population, but we believe that they cannot be applied in the context of mechanistic modeling. As far as we can see, there are no established standards for validating computational models in systems biology, but we offer several lines of arguments for the validity of our modeling results: 1) the mechanistic plausibility of the model, 2) the determination of its parameters by either direct measurement or by systematic fits to specific sets of experiments, 3) the consistency of the model with experimental measurements, 4) the consistency of the different types of experiments, i.e. SPR, in vitro and in-cell measurements can be reproduced using very similar parameter values. On top of that we include an additional supplementary figure (Supplementary Fig. 7) in the revised manuscript that shows that the parameters determined by fitting are well-constrained by the data and that the model predictions are robust to changes in parameter values.

Even more specific comments:

1. Explanation of the early part of the story.

a) FxFV, FXFV, and FIFV. Is FIFV an example FxFV, present in ORF45? A better construction is needed at line 126.

Response: “FIFP” was meant to refer to the ORF45 FxFP motifs. To avoid confusion, we removed “FIFP” from the text and figures and refer to this region as “FxFP” only.

b) The idea that the FxFV sequence binds to RSK-NTD was not anticipated. This needs to be discussed in the Results section in the paragraph starting at line 147.

Response: At Line 152 we explicitly discuss now that the binding of the N-terminal FxFP containing ORF45 region was indeed unexpected as revealed in the RSK(NTK)-ORF45(16-76) binary crystal structure:

“The binding of the N-terminal ORF45 segment to the RSK NTK was unexpected since this region is known to bind to activated ERK with its FxFP motif (and binds ERK in the ERK-RSK-ORF45 ternary complex). The fact that this N-terminal ORF45 region is also visible in the RSK(NTK)-ORF45(16-76) binary crystal structure is consistent with the increased binding affinity of longer ORF45 constructs.”

Based on our biochemical characterization regarding RSK-ORF45 binding it appears that the VF motif is an absolute requirement for binding. However, we also demonstrated that VF core motif containing short peptides mediate only micromolar binding, and not low nanomolar binding as measured for the long ORF45(16-76) construct. Based on our former experience this is not unusual for linear motif based protein-protein interactions. There are often additional fuzzy contacts - that may not be apparent in protein-peptide crystal structures - but they contribute to binding affinity. What is unusual in the RSK-ORF45 interaction is that apart from the VF sequence (e.g. F66) we could not identify any single amino acid N-terminal to the VF-core that would have a major impact on its own. We believe that the N-terminal ORF45 region with its FxFP motif makes only “fuzzy” contacts to RSK(NTK) and we captured a low energy conformation out of an ensemble of potential conformations in the binary RSK(NTK)-ORF45(16-76) crystal structure. (The VF core region is not influenced by crystal packing, while the N-terminal region may be locked into a specific conformation by forming additional extra contacts with a symmetry related complex.)

The synergism between the VF motif core and the auxiliary N-terminal region is intriguing and calls for further investigation. The N-terminal region varies in ORF45 homologs and this region could influence how ERK-RSK signaling is affected by different viruses.

c) The VF-motif is not mentioned in the introduction. First mentioned on Line 152.

Response: We describe the VF motif in the introduction at Line 47 in the revised version: “The N-terminal ORF45 region (~80 aa) has been shown to be required and sufficient for ERK and RSK activation. In particular, F66 located in a short relatively conserved region, referred to as VF motif henceforth, has been identified as the most critical for binding to and activation of RSK.”

2. Figures.

a) Fig. 1 is ok.

b) Fig. 2. Fig. 2A should be better annotated taking into account the multiple references to the VF segment throughout the paper. Fig. 2B, second panel is ok. All of the rest are inadequately labeled. The goal should be that someone with a passing knowledge of protein kinases can understand the view, and what is seen. In general, this problem can be fixed by labeling the secondary structure elements. In the discussion of this figure, panel 2B is not described. There are multiple interactions, and how the ORF45 weaves across the C-terminal domain of the kinase should be described. The fact that the FxFV motif is chaperoned in this 2-protein complex is very interesting and should be discussed, not just mentioned in passing as it is now.

Response:

The FxFP-motif and the VF-motif is now better annotated on Figure 2A. Secondary structural elements are indicated on B and C panels. Regarding the N-terminal ORF45 region as seen on panel C, please see our response at Remark 1b.

c) Fig. 3a Deuterium is shown is not described. Statistics? Cutoff values? The description in the methods section says for example on line 635 of the methods it says “proteins were mixed” (with no explanation of which proteins). In discussing the SAXS data, something should be said about the good Kratky plots presented in the Supplemental. There is no discussion of the data collection. There is no acknowledgement to the beam-line or beam line scientist or authorship. No discussion of alternative models and their fitting. Also, it is customary to report on the radius of gyration, the Dmax in addition to Goodness of Fit. There should be a much better discussion of what went into the model, since one of the major goals of this project should be to understand what is special about this ternary complex.

Response: HDX-MS data is presented now in far more detail and according to the standards of the field:

We included a full data analysis table (see Supplementary Table 2), provided a more extensive description on how samples were handled, which is now in an additional section of the Supplementary methods, and we give information on critical peptides as Source data related to Supplementary Table 2.

The Kratky-plot analysis is also better presented now. At Line 114: “Furthermore, small X-ray scattering (SAXS) analysis of RSK2 alone, the binary ERK2-RSK, and the ERK2-RSK2-ORF45 ternary complex showed that the proteins form a 1:1:1 stoichiometric complex, and dimensionless Kratky-plots indicated that the ternary complex has a compact domain arrangement (Supplementary Fig. 2b and Supplementary Fig. 3a,b,c).”

The support of SAXS beam line scientists are noted in the Acknowledgments: “The authors are thankful for the staff at BM29 beamline at ESRF and at the P12 beamline at EMBL-Hamburg (PETRA) for their help in SAXS data collection.”

R_g and D_{max} is written on Figure 3b. (Please note that these are the same values as in Supplementary Figure 2b in the table for the ternary complex, but the fit there was calculated with apoNTK and apoERK crystal structures without any ORF45 crystallographic model).

Regarding the description of what went into the ternary structural model for the SAXS analysis, please see Line 178 in the Results and the Methods at Line 613. We explicitly state now what part of the final ternary complex model came from crystal structures, how structured domains from these were treated as rigid bodies and what the flexible regions were and how these were handled in the CORAL simulation.

Fig. 4. OK and interesting binding data.

Fig. 5 and 6 See overall comments above.

3. Discussion. The discussion is over-long. The discussion of the model could state "with the number of docking interactions involved and the number of parameters, it is possible to build a model that fits opposing results, as described here in in vitro and in vivo outcomes of ORF45 signaling."

Response: We changed the first paragraph of the Discussion and state that “ORF45 binding blocks ERK mediated RSK activation in vitro but it activates RSK signaling in cells. Here we showed that a signaling network model comprised of the relevant kinase/phosphatase/substrate interactions and enzymatic reactions with realistic parameters can explain these opposing findings (see Fig. 6).” The Discussion was shortened by about one third of its former length.

REVIEWER COMMENTS

Reviewer #1 (Remarks to the Author):

The authors have responded to my suggestion of analyzing the effect of mutations in the KSHV or45 protein that disrupt binding to either RSK (F66A) or Erk (F32/34A) in the context of a mutated viral genome by (i) referring to a publication that included the former and (ii) preliminary data that contained the latter mutation. These results are in line with the authors' interpretation that the formation of a trimeric ORF45-RSK-ERK2 complex is required for the phosphorylation of RSK substrates and (in the case of the F66A mutation) productive viral replication. My comments have therefore been addressed in a satisfactory manner and I can understand the authors' decision not to include the above preliminary data in this manuscript.

Reviewer #2 (Remarks to the Author):

The authors have addressed all of my comments. I recommend this article for publication in Nature Communications.

Reviewer #3 (Remarks to the Author):

Overall, I like what the authors are attempting to do here. This is an ambitious, integrated, biophysical systems biology type study. The more the field moves in this direction, the better. And movement in this area will continue to advance this approach, so the authors do not need to be perfect.

I defer to the other reviewers on biophysical methods and interpretations in my review of the revision, as that is clearly more their area of expertise. This review of the revision focuses on the mathematical modeling.

The trends in the changes and updates made by the reviewers are positive with respect to better describing the model and more responsible interpretation of the model.

However, more changes are needed. Based on the changes in the revised manuscript and the authors' response to reviewers, this reviewer still has the impression that authors are giving much more credence to their mathematical model than it deserves.

I am supportive of this work – provided that the phrasing is substantially attenuated. Most readers will be less sophisticated than the authors, and the overly strong language will unnecessarily create obstacles to progress in the field.

Requested change 1:

Current text:

207 We created a quantitative mechanistic model based on the binary binding experiments and
208 obtained the binding rate constants that described the binary interactions between ERK2, RSK2,
or
209 ORF45 (k_{on} and k_{off-s} ; ERK2-RSK2, ERK2-ORF45 and RSK2-ORF45) (referred to as the
210 Biophysical 210 Model, see later).

Suggested change:

Rephrase as “We created a quantitative mechanistic model of the *in vitro* systems studied by SPR. We based our model on the binary binding experiments and obtained....”

Reason: Better clarify of what is modeled will help with communicating what is being done, and will better highlight the nature of the discrepancy.

Requested change 2:

Current text:

210 However, using the optimized parameters that fitted all binary
211 binding experimental data well, the SPR binding curves for the ppERK2-RSK2-ORF45 ternary
212 samples could not be reproduced, suggesting that ORF45 and ERK binding to RSK2 are not

213 independent (Supplementary Fig. 5).

Suggested change:

Add “possibly” or “likely” – I am okay with “likely” in this specific instance only because it is modeling well controlled and artificial in vitro SPR conditions. I am also okay with “likely” because the authors do a far better job communicating the difference between their “biochemical” and “network” models later in the text. However, some qualifier is still necessary.

Requested change 3:

Current text:

267 ORF45 inhibits ERK→RSK phosphorylation in vitro but it is an efficient activator of ERK-RSK

268 signaling in cells. We were intrigued by this discrepancy and given the presence of many interacting

269 molecules we decided to address the impact of ORF45 on the ERK signaling pathway by

270 mathematical modeling.

Reason this statement is problematic:

The phrasing “we decided to address the impact... by mathematical modeling” is very problematic as it implies far more power and value to the mathematical model than it deserves. What the authors now have is a model of an artificial system studied in SPR, and now they want to apply it to a more complicated in cell system. The fundamental assumptions of modeling in the SPR model (well-mixed, continuous variables fine, spatial factors can be ignored, stochastic factors not at play, rate constants measured in a very different set of conditions apply in this other set of conditions, etc.) can not necessarily be assumed to be at play in this situation. The authors address some of this in their later, rephrased, description of extending the model. However, this statement, at this portion of the narrative, endows the model with far more power than it deserves.

Suggested change:

Replace “address the impact” with “investigate the possible impact”

Requested changes 4 and 5:

Current text:

293 The Network Model - built based on a mechanistically sound

294 blueprint for kinase/phosphatase/substrate binding and catalytic reactions - correctly reproduced

295 ORF45 mediated up-regulation of ERK and RSK phosphorylation in cells (Fig. 6c).

Suggested change 4:

The phrase “mechanistically sound blueprint” is too strong of an interpretation/characterization. This contributes to the impression that the authors are heavily over interpreting the value of their model. All networks are subsets of larger networks. A subset of a large network does not necessarily have the same quantitative behavior as it displays in the larger network. Additionally, if spatial factors, etc., are at play the well-mixed approach used by the authors may not work. Etc. Thus, “ - built based on a mechanistically sound blueprint for kinase/phosphatase/substrate binding and catalytic reactions” implies that this is correct, and that it is sufficient. It is scientifically more responsible to rephrase as “ – which focuses on the submodule limited to kinase/phosphatase/substrate binding and catalytic reactions.”

Suggested change 5:

The phrase “the model correctly reproduced” is too strong of an interpretation. The word “correct” implies that the model is correct, which the authors readily state it is not (as the authors state it is a simplification/etc.). Please rephrase to “the model produces kinetic trajectories that are reasonably similar to the experimental observations”.

Requested changes 6 and 7 and 8:

Current text:

295 The viral

296 protein, instead of being a direct kinase activator, “activates” by protecting both kinases from

297 phosphatases in the cell⁷ .

Suggested change 6:

There is a space and an extraneous character in my text after the citation to reference 7. (Please check ms for other instances of this formatting glitch... it also appears in line 403 on my PDF.)

Suggested change 7:

I would add “the model suggests” before “the viral protein” – the authors haven’t shown this interpretation is true, they only show it is consistent with the model.

Suggested change 8:

Phosphatases appears to be mis-spelled.

Requested change 9 and 10:

Current text:

302 Since the Network Model was built after a comprehensive characterization of critical

303 pathway components inherited from the Biophysical Model, we could use it to address how key
304 mechanistic tenets of the model (e.g. the increased binding affinity of pERK (pE) to ORF45 (O),
305 $KD_{EO} > KD_{pER}$, or the nanomolar binding affinity between RSK and ORF45, $KD_{OR} \sim 1$ nM)
306 would affect the overall outcome (Fig. 6d).

Suggested change 9 :

The definition of “comprehensive” is “completed and including everything that is necessary”. Thus, the phrase “comprehensive characterization” is not accurate here, as the authors did not comprehensively characterize the pathway for modeling. This is the exact type of language that is so problematic in this manuscript. Despite the original reviews, the authors still seem to be overinterpreting their model... as this phrasing was NEW to the revised manuscript. The phrasing “extensive characterization” is still generous, but would be more appropriate and less problematic.

Suggested change 10:

“we could use it to address” needs to be restated as ‘we could use it to investigate”

Reviewer #4 (Remarks to the Author):

In this re-review, I found the paper much more accessible with many small errors in the figure labels and text fixed.

There are still relatively minor problems, especially with the description of Figure 3, the SAXS data on the ternary complex. The origin of the SAXS model really should be discussed better. It seems they have enough data to build this model with a crystal structure of the ERK-RSK-CTK and deuterium exchange protection to pull in the RSK2-NTK, but they have not discussed what was done to build the model, and what alternative models were tested. The rationale for putting disordered links into the “SAXS” model was not given.

Concerning their kinetic modeling, this is now improved.

I am surprised that they did not make more of the fact that conformational changes are required to form the ternary complex away from what is observed in the RSK2-ORF45 complex. This fact suggests an order of events, and explaining why RSK2 is not active in vitro in the presence of ORF45.

This paper probably will not be the last on the role of competition between substrates, activators, phosphatases, and hijackers like ORF45 for docking sites in ERK2 and other kinases.

Please find our point-by-point response in normal font after “**Response:**” to the reviewers’ comments (shown in full in *italics*).

Reviewer #1 (Remarks to the Author):

The authors have responded to my suggestion of analyzing the effect of mutations in the KSHV or45 protein that disrupt binding to either RSK (F66A) or Erk (F32/34A) in the context of a mutated viral genome by (i)referring to a publication that included the former and (ii) preliminary data that contained the latter mutation. These results are in line with the authors' interpretation that the formation of a trimeric ORF45-RSK-ERK2 complex is required for the phosphorylation of RSK substrates and (in the case of the F66A mutation) productive viral replication. My comments have therefore been addressed in a satisfactory manner and I can understand the authors' decision not to include the above preliminary data in this manuscript.

Response: We thank the reviewer for the earlier comments and the appreciation of our work.

Reviewer #2 (Remarks to the Author):

The authors have addressed all of my comments. I recommend this article for publication in Nature Communications.

Response: We thank the reviewer for the earlier comments and the appreciation of our work.

Reviewer #3 (Remarks to the Author):

Overall, I like what the authors are attempting to do here. This is an ambitious, integrated, biophysical systems biology type study. The more the field moves in this direction, the better. And movement in this area will continue to advance this approach, so the authors do not need to be perfect.

I defer to the other reviewers on biophysical methods and interpretations in my review of the revision, as that is clearly more their area of expertise. This review of the revision focuses on the mathematical modeling.

The trends in the changes and updates made by the reviewers are positive with respect to better describing the model and more responsible interpretation of the model.

However, more changes are needed. Based on the changes in the revised manuscript and the authors’ response to reviewers, this reviewer still has the impression that authors are giving much more credence to their mathematical model than it deserves.

I am supportive of this work – provided that the phrasing is substantially attenuated. Most readers will be less sophisticated than the authors, and the overly strong language will unnecessarily create obstacles to progress in the field.

Requested change 1:

Current text:

207 We created a quantitative mechanistic model based on the binary binding experiments and

208 obtained the binding rate constants that described the binary interactions between ERK2, RSK2, or ORF45 (kon and koff-s; ERK2-RSK2, ERK2-ORF45 and RSK2-ORF45) (referred to as the Biophysical Model, see later).

Suggested change:

Rephrase as “We created a quantitative mechanistic model of the in vitro systems studied by SPR. We based our model on the binary binding experiments and obtained....”

Reason: Better clarify of what is modeled will help with communicating what is being done, and will better highlight the nature of the discrepancy.

Requested change 2:

Current text:

210 However, using the optimized parameters that fitted all binary binding experimental data well, the SPR binding curves for the ppERK2-RSK2-ORF45 ternary samples could not be reproduced, suggesting that ORF45 and ERK binding to RSK2 are not independent (Supplementary Fig. 5).

Suggested change:

Add “possibly” or “likely” – I am okay with “likely” in this specific instance only because it is modeling well controlled and artificial in vitro SPR conditions. I am also okay with “likely” because the authors do a far better job communicating the difference between their “biochemical” and “network” models later in the text. However, some qualifier is still necessary.

Requested change 3:

Current text:

267 ORF45 inhibits ERK→RSK phosphorylation in vitro but it is an efficient activator of ERK-RSK signaling in cells. We were intrigued by this discrepancy and given the presence of many interacting molecules we decided to address the impact of ORF45 on the ERK signaling pathway by mathematical modeling.

Reason this statement is problematic:

The phrasing “we decided to address the impact... by mathematical modeling” is very problematic as it implies far more power and value to the mathematical model than it deserves. What the authors now have is a model of an artificial system studied in SPR, and now they want to apply it to a more complicated in cell system. The fundamental assumptions of modeling in the SPR model (well-mixed, continuous variables fine, spatial factors can be ignored, stochastic factors not at play, rate constants measured in a very different set of conditions apply in this other set of conditions, etc.) can not necessarily be assumed to be at play in this situation. The authors address some of this in their later, rephrased, description of extending the model. However, this statement, at this portion of the narrative, endows the model with far more power than it deserves.

Suggested change:

Replace “address the impact” with “investigate the possible impact”

Requested changes 4 and 5:

Current text:

293 The Network Model - built based on a mechanistically sound blueprint for kinase/phosphatase/substrate binding and catalytic reactions - correctly reproduced

295 ORF45 mediated up-regulation of ERK and RSK phosphorylation in cells (Fig. 6c).

Suggested change 4:

The phrase “mechanistically sound blueprint” is too strong of an interpretation/characterization. This contributes to the impression that the authors are heavily over interpreting the value of their

model. All networks are subsets of larger networks. A subset of a large network does not necessarily have the same quantitative behavior as it displays in the larger network. Additionally, if spatial factors, etc., are at play the well-mixed approach used by the authors may not work. Etc. Thus, “ - built based on a mechanistically sound blueprint for kinase/phosphatase/substrate binding and catalytic reactions” implies that this is correct, and that it is sufficient. It is scientifically more responsible to rephrase as “ - which focuses on the submodule limited to kinase/phosphatase/substrate binding and catalytic reactions.”

Suggested change 5:

The phrase “the model correctly reproduced” is too strong of an interpretation. The word “correct” implies that the model is correct, which the authors readily state it is not (as the authors state it is a simplification/etc.). Please rephrase to “the model produces kinetic trajectories that are reasonably similar to the experimental observations”.

Requested changes 6 and 7 and 8:

Current text:

295 The viral

*296 protein, instead of being a direct kinase activator, “activates” by protecting both kinases from
297 phosphatases in the cell⁷.*

Suggested change 6:

There is a space and an extraneous character in my text after the citation to reference 7. (Please check ms for other instances of this formatting glitch... it also appears in line 403 on my PDF.)

Suggested change 7:

I would add “the model suggests” before “the viral protein” – the authors haven’t shown this interpretation is true, they only show it is consistent with the model.

Suggested change 8:

Phosphatases appears to be mis-spelled.

Requested change 9 and 10:

Current text:

*302 Since the Network Model was built after a comprehensive characterization of critical
303 pathway components inherited from the Biophysical Model, we could use it to address how key
304 mechanistic tenets of the model (e.g. the increased binding affinity of pERK (pE) to ORF45 (O),
305 $KD_{EO} > KD_{pER}$, or the nanomolar binding affinity between RSK and ORF45, $KD_{OR} \sim 1$
nM)*

306 would affect the overall outcome (Fig. 6d).

Suggested change 9 :

The definition of “comprehensive” is “completed and including everything that is necessary”. Thus, the phrase “comprehensive characterization” is not accurate here, as the authors did not comprehensively characterize the pathway for modeling. This is the exact type of language that is so problematic in this manuscript. Despite the original reviews, the authors still seem to be overinterpreting their model... as this phrasing was NEW to the revised manuscript. The phrasing “extensive characterization” is still generous, but would be more appropriate and less problematic.

Suggested change 10:

“we could use it to address” needs to be restated as ‘we could use it to investigate’

Response: We thank the reviewer for the constructive criticism and for the appreciation of what we tried to do in this work: indeed, we aimed for a “biophysical systems biology type study”. We fixed the typos and removed the extraneous characters due to a formatting glitch. We improved the language regarding the modeling aspects exactly as suggested under all Suggested changes (1 through 10).

Reviewer #4 (Remarks to the Author):

In this re-review, I found the paper much more accessible with many small errors in the figure labels and text fixed.

There are still relatively minor problems, especially with the description of Figure 3, the SAXS data on the ternary complex. The origin of the SAXS model really should be discussed better. It seems they have enough data to build this model with a crystal structure of the ERK-RSK-CTK and deuterium exchange protection to pull in the RSK2-NTK, but they have not discussed what was done to build the model, and what alternative models were tested. The rationale for putting disordered links into the "SAXS" model was not given.

Response: We improved the description of the SAXS model by being more specific on how we constructed the final model and compared it to another possible model in the first revision. We further revised the text and describe what the overall rationale in the construction of this model was, how disordered linkers between the globular domains were used in the final model, and how the latter was selected. The concrete text that we introduced into the manuscript in the Material and Methods section under the subheading entitled "*HDX-MS, SAXS analysis and structural modeling*" is the following and it starts at Line 613:

"The overall logic in the construction of the ternary complex model was the following: we took the crystal structures of three known binary complexes - ERK2-CTK(RSK1), ORF45-NTK(RSK2), and ORF45-ERK2 - and used the SAXS data to guide the quaternary structure refinement of these rigid bodies connected with flexible linkers (since the NTK is connected to the CTK and the C-terminal VF region of ORF45 is linked to its N-terminal FxFP region flexibly). We assumed that the binary contacts for ERK2-CTK, ORF45(VF)-NTK, ORF45(FxFP) binding will be the same as in the ternary complex observed in their binary crystal structures and that the linker between the NTK and CTK of RSK2 links these two globular domains flexibly. Since the binary crystal structures do not have information on all residues because some of the protein regions are likely flexible/disordered, these terminal and linker regions contribute to scattering notwithstanding, we introduced them as flexible regions with dummy atoms and allowed them to move freely. The disordered linkers provided sterical restraints on the possible arrangements of the globular binary complexes in SAXS data driven structural modeling and helped to find the best quaternary arrangement between the three globular kinase domains, selected based on the model with the best χ^2 given by CORAL. The binary contacts withing the ternary complex - originally taken from the binary crystal structures - were validated by HDX-MS experiments (see Fig. 3a). The real challenge was to find out how the NTK as a globular domain may be incorporated into the ternary complex given that its association with ORF45 is mediated by the C-terminal VF-motif region of the latter but its N-terminal FxFP region binds to ERK2 (which binds to the CTK that is flexibly linked to NTK). The final ternary complex model was consistent with the observation that the NTK's substrate binding pocket and active site are accessible for substrate binding and phosphorylation (see Fig. 5b)."

In addition, per editorial request, we also included the SAXS data collection table as Supplementary Table 4.

Concerning their kinetic modeling, this is now improved.

I am surprised that they did not make more of the fact that conformational changes are required to form the ternary complex away from what is observed in the RSK2-ORF45 complex. This fact

suggests an order of events, and explaining why RSK2 is not active in vitro in the presence of ORF45.

Response: We did not mean to suggest that RSK2 is NOT active in the presence of ORF45 in vitro. Figure 5a shows that RSK2 phosphorylation is what is blocked by ORF45 in vitro, and these experiments did not address RSK2 activity on its substrates. RSK mediated phosphorylation on this figure is tested only on panel b, but this shows that RSK mediated substrate phosphorylation is unaffected in the context of the ternary complex. Moreover, ORF45 has no impact on the phosphorylation of the EIF4B substrate by pre-activated RSK2 in vitro (see Figure 7D, lowest panel). Note that EIF4B does not have a VF-motif and thus its phosphorylation is independent from VF-motif docking and its phosphorylation is mitigated only by the enzyme's substrate binding pocket. It is true that the substrate binding pocket of inactive NTK seems to be blocked in the RSK2(NTK)-ORF45 binary crystal structure, apart from the VF-motif docking groove, however, we believe that the former is functionally not relevant as we discussed this in our response in the first revision. Therefore, we do not think that there is an interesting order of events that requires the release of the NTK's substrate binding pocket from ORF45 mediated inhibition, as structurally this may seem to be needed based on the binary NTK-ORF45 crystal structure, because functional data do not support such a scenario. We learned that crystal structures of complexes with flexible regions need to be looked at with extra skepticism and apparent projections from these structures need to be backed up by functional data. Naturally, we were also enthusiastic about the binding of the N-terminal ORF45 region in the NTK's substrate binding pocket at first, but later experiments did not confirm its relevance in RSK mediated substrate phosphorylation.

This paper probably will not be the last on the role of competition between substrates, activators, phosphatases, and hijackers like ORF45 for docking sites in ERK2 and other kinases.

Response: We thank the reviewer for the critical comments and pointing out some flaws in the original submission, which we are hoping to have addressed and fixed after this revision. We agree that there will likely be more papers in the future on addressing the qualitative aspects of competition between different partners of protein kinases, since this provides an opportunity not only for pathogenic hijackers, but also for us – through the use of specific drugs – to modulate the activity of protein kinases in a more specific manner.